# Lessons from implementation research on community management of Possible Serious Bacterial Infection (PSBI) in young infants (0-59 days), when the referral is not feasible in Palwal district of Haryana, India

Rupak Mukhopadhyay[1]°, Narendra Kumar Arora[1]°*, Pradeep Kumar Sharma[2], Suresh Dalpath[2], Priya Limbu[1], Geetanjali Kataria[1], Rakesh Kumar Singh[1], Ramesh Poluru[1], Yogesh Malik[2], Ajay Khera[3], P. K. Prabhakar[3], Saket Kumar[2], Rakesh Gupta[2], Harish Chellani[4], Kailash Chander Aggarwal[4], Ratan Gupta[4], Sugandha Arya[4], Samira Aboubaker[5], Rajiv Bahl[6], Yasir Bin Nisar[6], Shamim Ahmad Qazi[5]

1 The INCLEN Trust International, New Delhi, India, 2 Department of Health, Government of Haryana, Palwal/Chandigarh, India, 3 Ministry of Health and Family Welfare, Government of India, New Delhi, India, 4 Department of Pediatrics, Vardhaman Mahavir Medical College & Safdarjung Hospital, New Delhi, India, 5 Retired World Health Organization Staff, Geneva, Switzerland, 6 Department of Maternal, Newborn, Child and Adolescent Health and Ageing, World Health Organization, Geneva, Switzerland

° These authors contributed equally to this work.
* nkarora@inclentrust.org

## Abstract

### Background

Neonatal sepsis is a major cause of death in India, which needs hospital management but many families cannot access hospitals. The World Health Organization and the Government of India developed a guideline to manage possible serious bacterial infection (PSBI) when a referral is not feasible. We implemented this guideline to achieve high coverage of treatment of PSBI with low mortality.

### Methodology

The implementation research study was conducted in over 50 villages of Palwal district, Haryana during August 2017-March 2019 and covered a population of 199143. Policy dialogue with central, state and district health authorities was held before initiation of the study. A baseline assessment of the barriers in the implementation of the PSBI intervention was conducted. The intervention was implemented in the program setting. The research team collected data throughout and also co-participated in the implementation of the intervention for the first six months to identify bottlenecks in the health system and at the community level. RE-AIM framework was utilized to document implementation strategies of PSBI management guideline. Implementation strategies by the district technical support unit (TSU) included: (i) empower mothers and families through social mobilization to improve care-seeking of sick young infants 0–59 days of age, (ii) build capacity through training and build

**Data Availability Statement:** The data underlying the results presented in the study are available from Brig (Rtd.) V. K. Panday (Email: vk. panday@inclentrust.org; Phone - +91-8607023300).

**Funding:** The study was supported by a grant from Bill and Melinda Gates Foundation (OPP1114815) to the Department of Maternal, Newborn, Child and Adolescent Health, World Health Organization, Geneva, Switzerland. The funder had no role in study design, data collection and analysis, decision to publish, or preparation of the manuscript.

**Competing interests:** The authors have declared that no competing interest exist. Rajiv Bahl and Yasir Bin Nisar are staff members of the World Health Organization. The expressed views and opinions do not necessarily express the policies of the World Health Organization.

confidence through technical support of health staff at primary health centers (PHC), community health centers (CHC) and sub-centers to manage young infants with PSBI signs and (iii) improve performance of accredited social health activists (ASHAs).

## Findings

A total of 370 young infants with signs of PSBI were identified and managed in 5270 live births. Treatment coverage was 70% assuming that 10% of live births would have PSBI within the first two months of life. Mothers identified 87.6% (324/370) of PSBI cases. PHCs and CHCs became functional and managed 150 (40%) sick young infants with PSBI. Twenty four young infants (7-59days) who had only fast breathing were treated with oral amoxicillin without a referral. Referral to a hospital was refused by 126 (84%); 119 had clinical severe infection (CSI), one 0–6 days old had fast breathing and six had critical illness (CI). Of 119 CSI cases managed on outpatient injection gentamicin and oral amoxicillin, 116 (96.7%) recovered, 55 (45.8%) received all seven gentamicin injections and only one died. All 7–59 day old infants with fast breathing recovered, 23 on outpatient oral amoxicillin treatment; and 19 (79%) received all doses. Of 65 infants managed at either district or tertiary hospital, two (3.1%) died, rest recovered. Private providers managed 155 (41.9%) PSBI cases, all except one recovered, but sub-classification and treatment were unknown. Sub-centers could not be activated to manage PSBI.

## Conclusion

The study demonstrated resolution of implementation bottlenecks with existing resources, activated PHCs and CHCs to manage CSI and fast breathers (7–59 day old) on an outpatient basis with low mortality when a referral was not feasible. TSU was instrumental in these achievements. We established the effectiveness of oral amoxicillin alone in 7–59 days old fast breathers and recommend a review of the current national policy.

## Introduction

Over half a million neonates died in 2019 in India and 33% of these deaths were due to one or more infectious causes [1]. Around 10% -13% of newborns and infants below two months develop symptoms and signs suggestive of possible serious bacterial infection (PSBI) [2–5]. Although the recommended treatment for possible serious bacterial infection (PSBI) is hospitalization [6, 7], referral to higher facilities and hospital admission remains challenging in several low and middle-income countries [8–11]. A verbal autopsy study from the study area (then called Mewat district) showed that 22.3% of neonatal deaths were due to sepsis and infections; and for 52.6% of these neonates, the families did not seek care outside their homes before death [12]. In the study in Delhi slums [8], only 24% of the PSBI infants complied with hospital referral; the reasons for non-compliance were–child not perceived to be ill enough for hospitalization by the family, no one to accompany the mother or care for other siblings, waiting for the response to medicines advised mostly by the unqualified local practitioners, sought medicine from other physicians, unpleasant past experiences of the hospital, and trial of home remedies. In the Bangladesh study [13], 28.5% of PSBI infants died when left untreated or treated by unqualified health providers.

Large randomized controlled trials from African and Asian countries demonstrated efficacy of simplified antibiotic therapy in outpatient setting [3–5, 14]. The World Health Organization developed a guideline for the management of PSBI in young infants up to two months of age with simplified antibiotic regimens when referral to a hospital was not feasible [15]. The government of India (GOI) had released a guideline for auxiliary nurse midwives (ANMs) in 2014 to manage PSBI on an outpatient basis with injection gentamicin and oral amoxicillin when a referral to a hospital was not feasible (revised in 2017) [16, 17]. It remained largely unimplemented due to the lack of understanding of operational and contextual bottlenecks for implementation of the guidelines in most parts of the country. In a resource-poor setting careseeking for young infants remains challenging due to distance to hospital, accessibility, affordability, time cost, wage loss, concern about the quality of care or attitude of the health workers, and cultural issues [8–11, 18, 19]. In the study district (Palwal, Haryana-India), during 2016–17 and 2017–18, only 1.1% and 1.7% of all live births respectively were identified as having any sickness including PSBI by the health workers [20]. The referral compliance and outcome were not known. The objectives of this implementation research were to understand the programmatic bottlenecks, determine the feasibility and acceptability of contextually modified implementation strategies and increase access to PSBI treatment using WHO PSBI management guideline when the referral is not feasible [15]. The implementation was embedded within the existing national program strategies for newborn–*Janani Suraksha Yojna* (Safe Motherhood Program) [21], Home Based Newborn Care Program (HBNC) [22] and Integrated Management of Newborn and Childhood Illness (IMNCI) [23].

The implementation study was part of a WHO-Government of India initiative at four sites in India (Himachal Pradesh, Maharashtra, Uttar Pradesh and Haryana).

## Methodology

### Study population and study setting

The implementation research was conducted in Palwal district of Haryana (India) between August 2017 and March 2019, at INCLEN-SOMAARTH DDESS (Demographic, Developmental and Environmental Surveillance Site) (www.somaarth.org), comprising of a population of 199,143 in 50 villages of three administrative blocks (Hathin, Hodal and, Palwal) (Fig 1). In the Haryana Vision 2030 document [24], Palwal has been ranked lowest in the human development index and fares poorly across all three indicators of human development, health, education and per capita income. Palwal district is one of the aspirational districts of the Government of India (i.e. with amongst lowest development indicators). The crude birth rate of the district was 26/1000 (compared to 20.3 in Haryana and 20.4 in National) in 2017 [25]. The infant mortality rate in Palwal was 35/1000 live birth (Haryana-32.8 /1000; India-41/1000) and the neonatal mortality rate was 21/1000 live birth (Haryana 22.1 /1000; India 30/1000) [26, 27]. According to the HMIS (health management information system) [28], Palwal district recorded 24046 deliveries during the year 2018–2019; 77% (18615) were institutional and the remaining (23%; 5431) occurred in homes. Only 17% (946/5431) of the home deliveries were attended by any skilled birth attendant. Furthermore, a verbal autopsy study from the same area indicated that 25% of neonatal deaths were due to sepsis [12]. In 50% of death cases, the neonates were not taken to any health facility before death. The actual figure of newborns and young infants with PSBI who were not accessing a health facility might be even higher. The preferred choice of care-seeking for neonatal illness in the area has been unqualified village practitioners followed by government hospitals and private qualified practitioners [29]. In a recent study on community management of pneumonia from the same study area 78% of sick

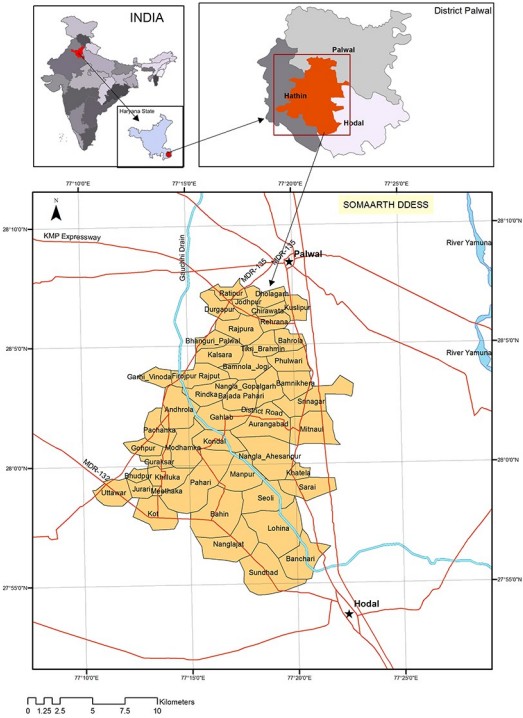

**Fig 1. Study area- INCLEN-SOMAARTH DDESS (Demographic Developmental and Environmental Surveillance Site) Palwal, Haryana (India).**

infants and children visited private care providers (qualified providers, chemist, traditional healers) [30].

## Health infrastructure

Primary healthcare services in the study were provided through a network of 18 Sub-Centers (SCs), three Primary Health Centers (PHCs) and three Community Health Centers (CHCs), supported by a referral facility at the district hospital. A brief description of health infrastructure and available manpower during the study period is provided (Table 1). None of the PHCs and CHCs had a functional newborn stabilization unit. The referral and in-patient facility for newborn [18 bedded -sick-newborn care unit (SNCU)] was located in the district hospital at Palwal City, almost 30KMs away from the farthest PHC of the study area.

## Study design

RE-AIM framework [31] was adopted to evaluate the implementation of PSBI guidelines in the study (Fig 2). The district health system led the implementation of the PSBI management guideline (intervention). The research team co-participated in the implementation for the first six months to get a better insight into barriers and bottlenecks and later monitored the processes and impact of the implementation strategy for another 12 months.

## Implementation strategy

To contextualize our implementation of the PSBI management intervention package, following discrete ERIC classification implementation strategies [32] were followed.

**Table 1. Government district health facility framework and description of facilities in the context of PSBI management.**

| Sl. no | Name of the Facility/ Institution | Expected strength of health personnel | Type of care provided | Population covered | Number of health facilities in the study area | Number of health personnel in position during the study period for neonatal care |
|---|---|---|---|---|---|---|
| 1 | **Accredited Social Health Activist (ASHA)** | Sanctioned positions for the area-172 | Social mobilizer for promoting maternal and child health: ante-natal care; deliveries; immunization and sickness identification, counseling, door-to-door surveys for specific service recipients | @One ASHA for approximately 1000 rural population | Working from home | 172 |
| 2 | **Sub-Centre (SC)** | 1. Two ANMs one male multi-purpose health worker (MPW) | Lower most health service delivery facility; provides basic antenatal care, immunization and treatment for minor illnesses | 3000–5000 | 18 | ANM: 29 Male Multi Purpose Health worker -3 |
| 3 | **Primary Health Centre (PHC)** | 1. Two qualified doctors (1-allopathic-MBBS and 1-*Ayush* Traditional medicine physician) One staff nurse 2. One ANM Other staff not directly involved with neonatal care: 1 pharmacist; 1 health educator and 2 health assistants | PHC- first port of call to a qualified public sector doctor in rural areas for the sick and those who directly report or referred from Sub-Centres *(5–6 SC fall under each PHC)* for curative, preventive and promotive health care. 4–6 Bedded facility | 20000–30000 | 3 | **PHC 1: Nagaljaat** Medical Officer: 0 Nurse: 1 ANM: 2 **PHC 2: Kot** Medical Officer: 2 Nurse: 0 ANM:1 **PHC 3: Uttawar** Medical Officer: 2 Nurse: 0 ANM:1 |
| 4 | **Community Health Centre (CHC)** | 1. Five medical specialists (i.e., general surgeon, physician, gynecologist, anesthetist and pediatrician) 2. Four general duty officers (1 dental surgeon, 3 general medical officers 3. Twenty one paramedical and support other staff. | First referral unit for SCs and PHCs; Facilities with one operation theater, X-ray machine, labour room and laboratory. Provides facilities for 24X7 obstetric care and specialist consultations Beds: 20–30 | 80000–120000 | 3 | **CHC 1: Aurangabad** Specialist: 0 Medical Officers: 4 Nurse: 1 ANM: 3 **CHC 2: Hodal** Specialist: 0 Medical Officers: 4 Nurse: 4 ANM: 4 **CHC 3: Hathin** Specialist: 0 Medical Officers: 5 Nurse: 4 ANM: 2 |
| 5 | **District Hospital (DH)** | Palwal district hospital is a 200 bedded hospital. The district hospital has 1. Thirty seven specialists (including 3 paediatricians) 2. One hundred and thirty two paramedical staffs (including 90 Staff nurse) | District Hospital is secondary referral level facility and provides comprehensive secondary health care services to the population in the district. District Hospital is expected to deliver Essential (Minimum Assured Services) and Desirable (which we should aspire to achieve) package of services. The services include OPD, indoor and emergency service. In addition, basic specialty services, newborn care, psychiatric services, physical medicine and rehabilitation services, accident and trauma services, dialysis services and anti-retroviral therapy. | Average population of a district varies between 15 to 30 million Palwal district hospital covers a population of 1 million | 1 | **Palwal Civil Hospital** Paediatrician:3 Medical officers:6 Nurse: 6 A deputy Chief Medical Officer [Deputy-CMO] supervised ANMs and ASHA coordinators [supervisors for ASHAs] Health administration in the district managed by Chief Medical Officer's Office |

*(Continued)*

Table 1. (Continued)

| Sl. no | Name of the Facility/ Institution | Expected strength of health personnel | Type of care provided | Population covered | Number of health facilities in the study area | Number of health personnel in position during the study period for neonatal care |
|---|---|---|---|---|---|---|
| 6 | **Special Newborn Care Unit (SNCU)** | Managed by an adequately trained pediatrician, doctors, staff nurses and support staff to provide 24x7 services. As per MOHFW, the expected staff strength for a SNCU should be 1. One doctor per 4 beds 2. One nurse per 1.5 beds (Palwal SNCU should have 5 doctors and 12 nurses) | A SNCU is established at the district hospital and sub-district hospitals to provide care for sick newborns, i.e., all type of neonatal care except assisted ventilation and major surgeries. It is a separate unit close to the labour room with 12 or more beds. Palwal DH- SNCU has 18 beds to manage sick newborns | >3000 to 20000 annual delivery at district /sub-district level | 1 | Same as mentioned in row 5 under the district hospital |

**1. Built coalitions.** The implementation study established a coalition at four levels. i). Ministry of Health and Family Welfare (MOHFW), Government of India. ii). State health department through Directorate of National Health Mission and obtained permission to undertake the study- deputy director of health agreed to actively participate in the functioning of the Technical Support Unit (TSU) established at the district level. iii). TSU oversaw the execution of the study with CMO (chief medical officer) as the chair who was also inducted as one

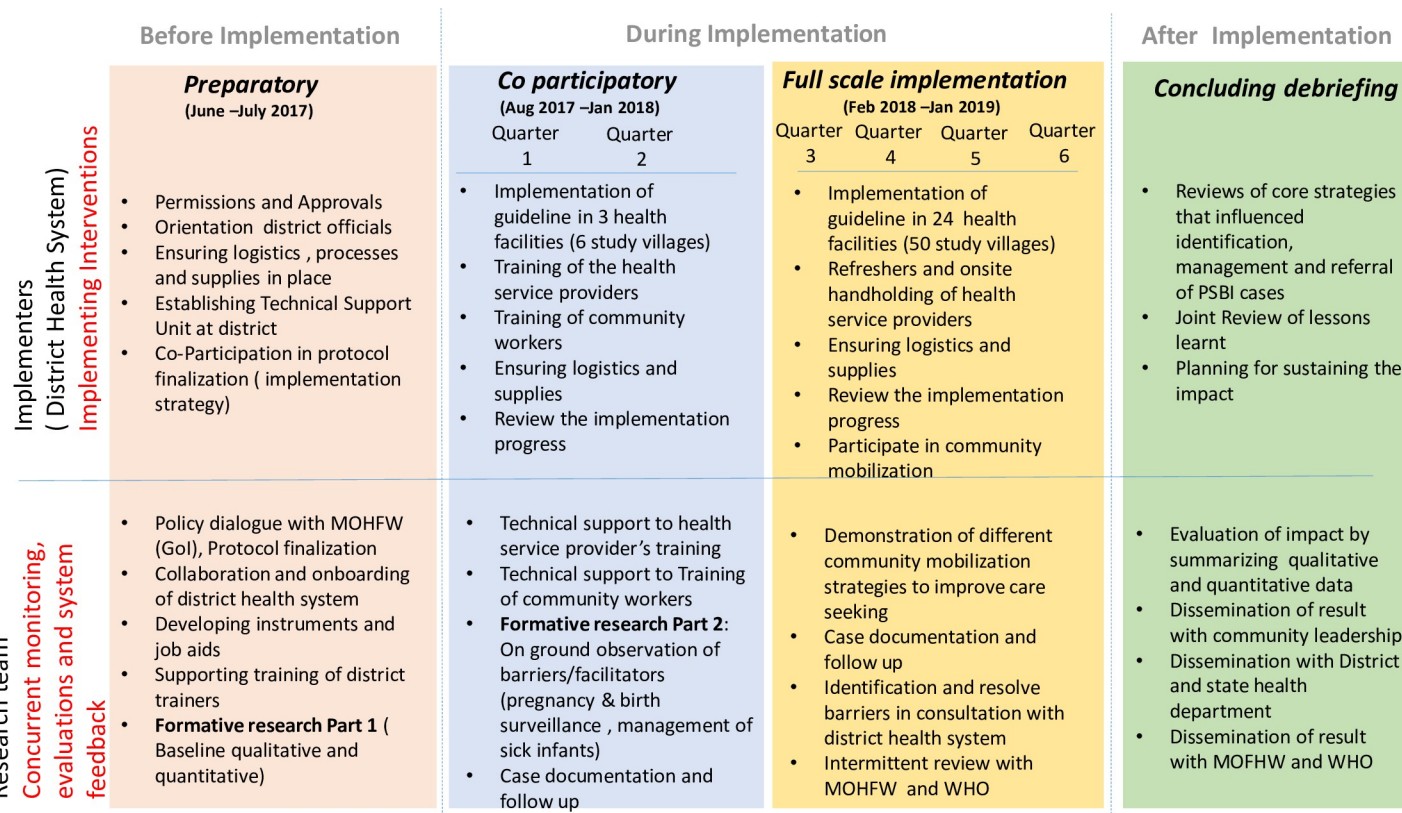

**Fig 2. Conceptual framework for implementation research (Modified RE-AIM framework).**

of the investigators–the research team was formally introduced to different functionaries of the health department in the study villages. iv). Series of meetings with Panchayat members (local self-government) to inform about the project objectives and to obtain community willingness to participate.

As part of the policy dialogue with both central and state ministries of health, the research team obtained permission for modification of the GOI PSBI management guideline in a national workshop (October 2016) attended by national and state program managers, independent experts and brought it in line with the WHO PSBI guideline specifically for the implementation research study (Table 2). The same group of stakeholders also participated in the

**Table 2. Differences between PSBI management guideline of Government of India (2014) and World Health Organization (2015).**

| SL no | Area | Specifics | |
|---|---|---|---|
| | | **Government of India Guideline (2014)[16]** | **WHO Guideline (2015)*[15]** |
| 1 | Danger signs to define PSBI | **Eleven signs** considered; the presence of one or more of these eleven signs defines PSBI. | Definition simplified by the presence of one or more of the only **seven signs**. Four signs were removed (nasal flaring, grunting, presence of pustules or big boil, blood in stool) |
| 2 | Simple fast breathing in 7–59 day old as a separate category and its treatment versus fast breathing in ≤ 6-day old infants | Fast breathing infants (≤59 days with or without other danger signs) considered PSBI and recommended to be treated as other PSBIs (Single daily dose of injectable gentamicin and twice-daily oral doses of amoxicillin for 7 days) if a referral is not possible | Young infant aged 7–59 days, presenting with only fast breathing and no other danger sign, defined as **Pneumonia.** Recommended treatment with twice-daily oral doses of amoxicillin for seven days and no injectable gentamicin on an outpatient basis. WHO guideline does not recommend referral to a hospital.<br><br>Young infant aged less than or equal to six days presenting with only fast breathing and no other danger signs, defined as **Severe Pneumonia,** and managed as other PSBI conditions. Recommended referral to a hospital. If the family refuses to accept the referral advice, treat with single daily IM gentamicin and twice-daily oral amoxicillin for 7 days on an outpatient basis. |
| 3 | Dosage of oral amoxicillin | Recommended - 25mg/kg/dose twice daily | Recommended - 50mg/kg/dose twice daily for 7 days |
| 4 | Dosage of intramuscular injection gentamicin | 5mg/kg/dose once daily for 7 days | 5mg/kg/dose once daily for 7 days |
| 5 | Reclassification when referral is not possible | No such reclassification is mentioned | If a referral is not possible, WHO guideline recommends following the reclassification of PSBI cases based on the presence of one or more following danger signs-<br>**SEVERE PNEUMONIA:** Please see above<br>**CLINICAL SEVERE INFECTION:** at least one sign of severe infection, i.e. not feeding well on observation, temperature 38˚C or more, temperature less than 35.5˚C, severe chest in-drawing, movement only when stimulated.<br>Recommended referral to a hospital. If the family refuses to accept the referral advice, treat with single daily IM gentamicin and twice-daily oral amoxicillin for 7 days on an outpatient basis.<br>**CRITICAL ILLNESS:** In a young infant, presence of any of the following signs: convulsions, unable to feed at all, no movement on stimulation, unable to cry, bulging fontanels, cyanosis<br>Recommended referral to a hospital. If the family refuses to accept the referral advice, treat with single daily intramuscular gentamicin and twice-daily injectable ampicillin until the referral is possible. |

*The WHO guideline was used for the implementation study

finalization of the research protocol. The research team provided technical support to the Haryana State NHM to modify the IMNCI recording form and PSBI case follow up card to manage and maintain follow-up records of PSBI infants treated in PHCs/CHCs.

**2. Centralize technical assistance.** A district Technical Support Unit (TSU) was constituted with the district CMO as the chairperson and representation from state directorate, district health administration, independent technical experts and the investigators from the research team. Terms of reference for the TSU included the following. i) To identify implementation bottlenecks based on the feedback from formative research. ii) To collate field experience of the district program managers and available PSBI implementation related evidence from other contexts particularly within the country. iii) To facilitate the development of implementation blueprint taking in to account the study context and its operational feasibility. A nodal officer was designated in the district health department to oversee the implementation, support participation of health personnel and monitor availability of supplies and logistics for the PSBI management, identify skill-building needs (training and re-orientation programs), and ensure regular and uninterrupted supplies of antibiotics (amoxicillin and gentamicin), equipment (thermometer and functional weighing scale) at all levels of health care along with the relevant stationery for the management of PSBI infants. TSU reviewed the implementation progress every two months for resolving operational issues and modified the strategy as required, and acted as a link between the state headquarters and the district health department and decided on mid-course corrections as and when required.

A quarterly progress report on case identification was prepared and shared with the district CMO office. Six-monthly progress and challenges were shared with the state health department and their help was sought whenever required.

**3. Work with educational institutions.** Three senior pediatricians and neonatologists from Vardhaman Mahavir Medical College and Safdarjung Hospital, Delhi, were taken on board for participating in policy dialogue with decision-makers, provide technical assistance to develop job- aids for field staff and messages for the community and ratify translation of IMNCI chart booklet in Hindi (local language). This manual has since become part of the national program document, and support training of medical officers and other health functionaries of the study area.

**4. Conduct educational outreach programs.** The outreach training was started alongside formative research and completed in a phased manner by January 2018 before the scale of the project to all the 50 villages. National trainers (n-4) along with pediatrician (n-1), medical officers (n-4), ASHA-trainers (n-2) facilitated 11workshops: six workshops for ASHAs (1-day orientation) and five for the remaining health staff (3-day training). Training format was modified to ensure that medical officers, ANMs and nurses physically evaluated and wrote prescriptions for the sick young infant and learnt counseling of mothers for home care. Interaction with ASHAs emphasized the importance and schedule of home visitation, essential newborn care, identification of danger signs and places where sick young infants could be managed.

**5. Assessment for readiness and identify barriers and facilitators.** A formative study was conducted at baseline between June 2017 and January 2018. The assessment was done in two parts.

*Formative research-part 1(June-August 2017).* Baseline assessment included four components. i) Facility assessment: WHO facility scoring tool [33] was adapted to assess readiness for the availability of PSBI related supplies, manpower, service delivery and mechanisms of monitoring implementation of PSBI guideline (DH-1, CHC-3, PHC-3, sub-centers-18). ii) ASHAs' perspectives about HBNC program, home visitation, sickness identification among young infants and care provision were obtained through knowledge attitude and practice (KAP)

survey (n- 60; at the rate of at least one ASHA per village). iii) Perspectives of recently delivered (<6 months) and lactating mothers: In-depth interviews (n-30 mothers) and KAP survey with 150 mothers (at the rate of 3 randomly selected mothers from every village) to document mothers' awareness of danger signs, care-seeking for their sick newborn and perceived value of home visitation by the ASHAs. iv) Towards the end of part 1 of the formative phase, both in-depth interviews and the quantitative survey indicated ground-level challenges about home visitation under HBNC by the ASHAs, assessment and management of young infants at the PHCs and CHCs and its influence on the care-seeking by the families for sick young infants. The respondents (surveys of mothers and ASHAs) had touched on these issues but in a restrained manner probably due to the sensitive nature of the problems. Non-formal interactions (NFI) [34–36] with primary care physicians working at CHCs and PHCs (n-5), pediatric specialists at district hospital (n-2), ASHAs (n-8), ANMs (n-5) and mothers (n-11) were undertaken to further explore these issues. Only senior investigators conducted NFIs which were considered critical to the success of the implementation research. No tape recording or notes were taken at the time of interaction but immediately after NFI, the summary of the interaction was noted down.

*Co-participatory formative research–part 2 (August 2017- January 2018).* The purpose was to get first-hand insight into challenges and barriers to implementing PSBI guideline beyond that obtained under Part 1 of the formative phase. The co-participant implementation phase was limited to six villages (population-29000); served by 71 ASHAs, two SCs, one CHC and the district hospital. During this period the research team also undertook house to house survey to determine the pregnancies missed by the health system (ASHAs), tracked them for deliveries and called up families with newborn to verify the ASHAs' scheduled visits (For every new delivery a schedule of expected home visits by the ASHA was made and research staff called up the families between 48 and 96 hours after the scheduled time to confirm the home visits.) At least one of the investigators observed the doctors assessing infants aged less than 59 days at the CHC (receiving patients from the above mentioned six villages) using a structured observation checklist (S1 Checklist). This component was part of a planned staged implantation scale-up of the study; after the end of the six months, the study was expanded to all 50 villages.

**6. Revision of the implementation blueprint after co-participatory formative research.** Post formative research–part 2 the TSU decided to take up two focused activities as part of the implementation plan: (i) a structured physician mentoring activity to assess and manage sick young infants at PHCs and CHCs with confidence; and (ii) mounting social mobilization activities to empower the communities, families and mothers to recognize sick young infants, seek appropriate care and information about the availability of treatment facilities for these infants at the PHCs and CHCs.

**7. Provide local technical assistance.** TSU prepared a roster for the district paediatricians to go to CHC/PHC for hand-holding and confidence building of the primary care physicians. A paediatrician-visit occurred once every three months in every CHC/PHC. After the initial detailed training of clinicians, on-site meetings for primary physicians were organized at three CHC levels. Four such meetings were organized during the study. In these meetings, doctors from the district hospital CMO office, technical experts from partner medical college (Safdarjung Hospital) and, district Pediatrician took part.

Also, the research team worked along with block ASHA coordinators with 12 groups of ASHAs (5–8 per group; total 77) from villages that had missed more than 40% of the pregnancies, to teach filling of HBNC forms and ANC registers. A social network group (WhatsApp ®) was formed for physicians to share case videos, sending queries to pediatricians and the platform was used to pre-empt the pediatrician in the district about case referrals.

Oral amoxicillin was available at all levels, but injection gentamicin was not present in the PHCs and CHCs of the whole district. The research team worked with TSU and the CMO, ensured the availability of logistics and commodities in all outreach facilities especially the availability of antibiotics, one ml syringe, newborn weighing scale and thermometers. The research team monitored the stock availability on monthly basis and the staffs were encouraged to send timely indents to avoid stock-outs. With the approval of the Child health division of Haryana National Health Mission, the IMNCI recording form, PSBI referral card and PSBI case-record registers were modified to align with the requirements of the project. These were then printed and distributed by the research team for the duration of the project.

**8. Increase demand through a structured social mobilization campaign.** Existing community outreach platforms were leveraged for empowering families. The mobilization activities focused on four aspects. i) Identification of danger signs by the families and mothers. ii) Awareness about ASHAs' home visitation schedule and her expected duties during home visitation. iii) Awareness of the availability of treatment facilities for sick young in PHCs and CHCs. iv) Dissemination of case-studies of successful recovery from illness after availing treatment from public health facilities.

To achieve the above objectives, multi-pronged social mobilization strategies were adapted–

I. Wall paintings (@ at least two per village): content included ASHAs home visitation schedule, the danger signs for the sick young infants and place where treatment was available.

II. Village Health and Nutrition Day (VHND) celebration and Village Health Sanitation Nutrition Council (VHSNC) meetings were activated as per the GOI guidelines. These platforms were used for communicating the schedule and purpose of home visitation by ASHAs under HBNC program. ASHAs used flipcharts and posters to create awareness of danger signs among pregnant women and other community members attending these meetings. ASHAs also shared experiences from PSBI cases who had recovered after receiving simplified treatment in PHCs and CHCs.

III. The district health system introduced sick infant danger sign counseling in ANC-clinic conducted under Pradhan Mantri Surakshit Matritva Abhiyan (PMSMA).

IV. The district health system added one page on the PSBI danger signs and schedule of ASHAs home visits in the Mother-Child Protection Card.

V. Adhikaar Yatra (Health Rights March)–Schoolchildren (studying in class 8 and 9), their teachers and area ASHAs organized 69 rallies (@ at least one rally per study village) to spread awareness on newborn care, danger signs of sickness and ASHAs' home visitations.

VI. The District administration of Palwal organized a *Super village Challenge*—a village level competition in the third and fourth implementation quarters. Villages in Palwal district competed to achieve water, sanitation and hygiene (WaSH) and newborn and maternal health targets. The initiative led to greater engagement of local self-government (Panchayat members) due to visibility and recognition of the local leadership.

VII. Community contact meetings: Four such meetings were organized by the research team to bring together the local leadership, teachers, religious leaders and the district health leadership to engage them in the development and roll-out of social mobilization activities, share the progress and success of the implementation of PSBI guideline in the area.

**9. Development and distribution of educational materials.** Job Aids—IMNCI chart booklets, IMNCI recording forms, and PSBI case follow up forms were modified by the research team in collaboration with technical partners, translated in Hindi and introduced

into the program for use by the ASHAs, ANMs and doctors. These were also distributed across all health facilities. Wallboards for treatment algorithms and antibiotic dosage were fixed in PHCs, CHCs and SCs. Pamphlets with key messages were distributed during school rallies.

### Data collection, management and analysis

The research data collection team comprised of Indian traditional medicine graduates with master's in public health, graduate social scientists and field workers with at least 10 years of schooling. The research personnel were trained as per their assigned tasks in a 3–5 days' workshop. They collected data for during the formative research; medical graduates interacted with medical officers in primary health facilities in the first two quarters, conducted post-treatment follow up (on 8th and 14th day) of PSBI diagnosis irrespective of the place where they were treated and abstracted administrative data from official documents of ASHAs, ANMs, PHCs, CHCs and district hospital. The data included: pregnancy, live birth, home visitation by ASHAs, sick young infants assessed and managed at different public sector health facilities, PSBI case classification formats, drugs administered, place of management (outpatient or in-patient), adherence, and follow up on day four and seven after initiating the treatment from all government health facilities. Written informed consents were taken from the families /stakeholders.

As part of the community mobilization and engagement activities, the families were asked to inform the research team telephonically if they chose to take their sick young infants to private providers (S1 Table). Data about obtaining treatment at private health facilities, clinical features at the time of seeking care and outcome was based on the recall by the mothers and families and available prescriptions. Information was obtained by the research team on the 8th and 14th day follow up post-initiation of treatment. The information about the case classification was not available for these cases. The processes and experiences of different aspects of implementation were recorded in the daily diaries of the study investigator and team leaders.

*Quantitative data* (KAP surveys of mothers and ASHAs, facility assessments, pregnancy and birth surveillance, assessment of sick young infants, treatment provided and outcome) was double entered in RedCap® double data module, for detecting inconsistencies and merged after validation. Data was exported to STATA® (Version 15.0) for analysis. Descriptive statistics were used to present coverage data, treatment adherence and cross-sectional survey data.

**Qualitative data.** Transcripts of recorded interviews (IDIs conducted during formative research with health service personnel, mothers and ASHAs) were prepared and complemented with field notes taken during non-formal interactions. All transcripts were entered in IQDAS (INCLEN Qualitative Data Analysis Software) [37]. Data were free-listed and key axial and selective codes were generated for analysis.

## Results/observations

Both qualitative and quantitative data were used to identify implementation bottlenecks.

Table 3 provides observations and contextualized implementations under three sections: ASHAs, health service providers (primary care physicians, ANMs, and nurses) and mothers and the community.

### Barriers to PSBI program

ASHAs were making home visits irregularly and incompletely. The mothers did not value these visits as they were unaware of the purpose of these visits. ASHAs didn't emphasize the danger signs of sick young infants and their record maintenance was poor. During the six months before the IR, no sick young infant was referred to the primary health facilities (PHCs

**Table 3. Implementation barriers and bottlenecks identified through formative research, application of contextualized strategies and their impact on PSBI program implementation.**

| Level | Observation made during formative phase—Challenges identified | Action taken and impact on the program implementation |
|---|---|---|
| **ASHAs (Accredited Social Health Activist) Performance** | 1. **Pregnancy Tracking:** House to house pregnancy survey was done by the research team in the 50 study villages. ASHAs did not share their records for five villages; in the remaining, ASHAs had missed 36% (857/2359) pregnancies in 45 villages. | • Block ASHA Coordinators and District Immunization Officer re-emphasized the quarterly door to door survey by ASHAs. However, no change occurred in the intensity of supervision and ASHAs continued to rely on their social network approach for pregnancy detection. |
| | 2. **Record maintenance:** Almost half (49.4%: 2605/5270) of the ASHAs refused to share their home visitation records with the research team citing incompleteness and not having permissions from their superiors. Most ASHAs maintained rough records before copying these in the official forms. HBNC (Home-based newborn care) forms and ANC (Ante-natal Clinics) records were incomplete/ partially filled. ASHAs mentioned that they did not know how to fill in ANC and HBNC visit recording forms. <br> 3. **Post-Natal Home Visitation:** Home visits were irregular, not as per schedule and ASHAs did not accomplish the tasks as expected. ASHAs did not emphasize the identification of danger signs to the mothers during the home visitations. | • Following administrative actions were taken: Twelve trainings and re-orientation sessions were conducted by ASHA supervisors and local ANMs with small batches of ASHAs (total-82; 5–9 ASHAs per batch) to reorient ASHAs on various HBNC and ANC indicators, filling reporting formats and counseling of mothers during home visitation, VHND (village health and nutrition day) and VHSNC (Village Health, Sanitation & Nutrition Committee) meetings. <br> • Record maintenance did not show any improvement; the practice of having rough record-keeping persisted <br> • The frequency and regularity of the home visitation did not improve (Table 4). |
| | 4. **Monitoring of Post-Natal home visitations:** The monitoring of ASHA's post-natal home visitation by ASHA Coordinators was practically absent. | • CMO convened a meeting of the ASHA coordinators in the second quarter of the study and thereafter were asked to report their supervisory tasks during monthly meetings. <br> • We could not verify the frequency and quality of monitoring visits by ASHA coordinators. |
| | 5. **Supplies of HBNC forms to ASHAs:** Availability was irregular during the formative phase | • On the direction of TSU, INCLEN ensured the availability of relevant stationery with all the ASHAs through the study period. |
| | 6. **Referral of Sick infants to PHCs (Primary Health Centers)/CHCs (Community Health Centers):** ASHAs were reluctant to refer sick young infants to CHCs and PHCs because 'doctors referred these infants without even assessing'. | • As the PHCs and CHCs became functional during the implementation research, ASHAs started referring sick young infants to these facilities. <br> • 11% of the PSBI infants were identified by ASHAs and were referred to primary care facilities compared to none in the six months before the launch of the study. |
| **Health Service providers (Primary care physicians in CHC/PHC, ANMs, Staff Nurses)** | 7. **Operationalization of sub-centers and ANMs:** Formative research revealed that ANMs did not consider themselves as care providers and did not have the confidence to assess and manage sick young infants. They looked up to physicians at PHCs and CHCs for guidance. | • As part of the research, joint training of ANMs, nurses and physicians was conducted, which gave the trainees opportunity to independently assess sick young infants and write simplified antibiotic prescriptions for them. <br> • ANMs could not be made confident to assess and manage sick young infants independently in the sub-centres. Only one ANM identified and managed two PSBI patients at one of the CHCs under the supervision of the medical officer. |
| | 8. **Management of Sick Young Infants by Primary care physicians:** <br> • PHC/CHC Doctors were reluctant to assess and treat sick young infants even after regular IMNCI / PSBI trainings. Sick infants were frequently referred by them without even preliminary assessment. <br> • Doctors were unwilling to fill up IMNCI recording forms <br> • Up to the co-implementation phase, doctors were hesitant to give injectable gentamicin | • As part of the study, the trainees were allowed to independently assess sick young infants and write simplified antibiotic prescriptions for them. <br> • IMNCI trained research team members accompanied sick infants in the first two implementation quarters to support medical officers in the assessment and management of sick young infants at PHCs/CHCs. This practice was gradually reduced and stopped in the third implementation quarter as they had become confident to assess and manage and appropriately refer sick young infants if required. <br> • TSU organized hand-holding visits by district paediatricians from the third implementation quarter onwards by rotation to the CHCs and PHCs. This was to re-affirm the appropriateness of the assessment and prescriptions for sick young infants by primary care physicians and enhance their confidence. Visiting paediatricians emphasized the rationality and effectiveness of prescribing simplified antibiotic therapy including injection gentamicin. <br> • From the second implementation quarter onward, the PSBI performance review of PHCs/CHCs became the standing agenda item of the CMO's quarterly review. <br> • Impact: Primary physicians at PHCs/CHCs were able to manage almost $1/3^{rd}$ of the PSBI cases in the first 2 quarters and thereafter ended the study with over 40% of PSBI cases being managed at primary care health facilities (Table 5). <br> • Doctors continued to be reluctant to fill IMNCI case classification forms throughout the study <br> • The prescription and administration of gentamicin improved as the study progressed |

*(Continued)*

**Table 3.** (Continued)

| Level | Observation made during formative phase—Challenges identified | Action taken and impact on the program implementation |
|---|---|---|
| | 9. **Referral of sick young infants to higher facilities:**<br>• The referral system was almost non-functional; doctors and other health staff from PHCs/CHCs referred sick young infants without referral notes and any written guidance about where to take the infant. None received any pre-referral treatment.<br>•PHC and CHC staff was usually not sure about the compliance by the families on suggested referral.<br>• Families faced difficulties due to little importance and priority accorded to the referral slips from the field, sub-centres or PHCs/CHCs at the district hospital and other higher-level facilities.<br>• The parents would many times decide to give up and return home without any treatment due to lack of clear guidance about the place of treatment for their sick infant, running from one facility to another or seeking care from private practitioners. | • On the advice of TSU, CMO office notified: (i) doctors at CHCs and PHCs were to inform paediatrician at the district hospital when referring any sick infants; (ii) referral note was made mandatory with a clear mention of the reason for referral and place of referral; and (iii) medical superintendent was apprised of the admission in SNCU and requested to conduct a monthly review of neonatal admissions.<br>• District hospital administration made available round the clock transport arrangement to regional medical college Nuh (50 KMs) and another tertiary hospital in Delhi (65 KMs).<br>• Despite these efforts, the referral system remained inadequate till the end of the research and could not be streamlined as desired. |
| | 10. **Governance and human resource management:** During the implementation period there was a frequent change in leadership (4 CMOs changed) and a strike by health staffs (once by ANMs and other paramedical personnel for 3–4 weeks) and ASHAs (twice for a total period of 6–7 weeks) | • The achievements of the implementation research were with available staff and no effort was made to re-deploy the staff in the study area. |
| | Also, there was a deficiency of doctors 23% (8/34-including district pediatricians); 45% nurse (13/29) and 24% (13/55) ANMs. All ASHA positions were filled. | |
| **Mothers and community** | 1. **Identification of Sick young Infants:** Formative research showed that 1/3rd (33.8%; 51/151) of the mothers could not mention even a single danger sign for their infants.<br>2. **Care-seeking behavior**: The majority of mothers and families did not realize the sickness of their young infants; this led to delayed care-seeking. Also, they were not sure about the appropriate health facilities for the treatment of their sick young infants.<br>3. **Perceptions of ASHA's Home Visitation:** Most families and mothers were neither aware of ASHA's home visitation schedule nor about its purpose. Therefore many mothers and families did not value post-natal home visitation made by AHSA.<br>4. **Opinion on Public Health Facilities:** Family had trust issues in the public health facilities particularly for their young ones at PHCs/CHCs; and or had a previous bad experience. | • Initial interaction between the research team and the families indicated that mothers could quickly learn to recognize the danger signs and were ready for prompt and timely care-seeking once they realized the baby was sick. The response was encouraging within the first quarter of the study (co-implementation phase).<br>• TSU advised implementing structured and multi-pronged, contextually relevant social mobilization activities, utilizing existing community platforms and institutions. Mobilization activities focused on four aspects. (a) Identification of danger signs by the families and mothers. (b) Awareness about ASHAs' home visitation schedule and her expected duties during home visitation. (c) Awareness of the availability of treatment facilities for sick young in PHCs and CHCs. (d) Dissemination of case-studies of successful recovery from illness after availing treatment from public health facilities. Through the study period, 87.6% (324/370) PSBI infants were brought by the mothers and families to health facilities |
| | | The strategy worked and 87% of the PSBI infants were brought by the mothers and families to health facilities. |

and CHCs); ASHAs complained that doctors at these places did not see the sick young infants below 6 months.

In general, there were some supply issues at the PHCs and CHCs. Gentamicin injections were not part of the essential drug list and hence supply was erratic. Stock-outs for thermometers, home visitation recording formats were frequently observed.

PHC and CHC medical officers were reluctant to assess and manage treat newborns even after trainings and diverted sick infants to unspecified higher centers (i.e., without specifically telling where to go) without assessment. ANMs did not consider themselves as treatment providers, were hesitant to independently treat sick newborns at sub-center. Their confidence was further shaken when medical officers also demonstrated their hesitancy for managing sick infants.

Very few mothers and families could recognize the initial symptoms of the sickness in their young infants and hence the delay in care-seeking. As part of the formative study, only 53% (80/151) of mothers mentioned one or more danger signs in the newborn (almost all of them talked about fever and some about the other features of sickness) without realizing the seriousness and outcome of illness in young infants. In addition to the factors mentioned above, the families lacked trust in the public health system for managing their sick young infants; these infants were commonly taken to private providers (both qualified and unqualified) during sickness.

### Identification of sick young infants and management

Total 370 young infants with signs of PSBI were identified and treated in 5270 live births (Fig 3). Treatment coverage was 70% assuming that on average 10% of live births would have PSBI within the first two months of life [2–5].

In the two years before the study, no PSBI cases were managed at either of the PHCs and CHCs. TSU suggested a structured handholding and mentoring exercise for the doctors working in the CHCs and PHCs (Table 3), which resulted in 31.6% (12/38) of the PSBI cases evaluated at PHCs and CHCs in the first quarter of the study (Table 4). During this period, no sick infants were identified or managed at sub-centers. In the majority of PSBI cases (87.6%; 324/370) families sought care by themselves from various types of health facilities and 11.4% (42/370) were identified and mobilized by ASHAs (Table 4).

### ASHAs performance

During the first four quarters of the study, 24 out of 322 (7.5%) of the PSBI infants were identified by ASHAs compared to none in the six months before the IR. We triangulated the administrative visitation records submitted by ASHAs to the health department with information obtained from the families through the telephone calls made by the research teams between August 2017 and July 2018 when 3254 deliveries had occurred in the study area. The research team made 16,977 calls during this period. The calls were made within 48 hours after the seven government recommended post-natal home visits (days 1, 3, 7, 14, 21, 28, 42) from 2001 mother-newborn duos and partially from another 487 mother-duos. ASHAs shared information about home visitation for only 1475 (1475/3254; 45.3%) mother-newborn duos, reporting that incomplete home visit records were not shared. Visit wise information abstracted from ASHAs official record and that based on the telephone call by the research staff are provided in Table 5. The families reported that 17% (336/2001) of the mother-newborn had no post-natal home visitation by the ASHAs compared to ASHAs data showing at least some visits for every mother-child duo.

### Referred out of and referred into PHC/CHCs

Only 18% cases (31/174) accepted a referral for hospitalization: critical illness (CI)-6, clinical severe infection (CSI) -23, and severe pneumonia (fast breathing in 0–6 days old) -2. A pre-referral dosage of injection gentamicin was given in 71% (22/31) of referred cases (Fig 3).

### Place of treatment

Finally 40.5% (150/370) were treated at PHCs/CHCs, 17.6% (65/370) at district hospital (n-60) and 5 infants at tertiary care facilities; whereas the remaining 42% were managed at private health facilities (Table 4). Among 150 cases treated at primary care facilities,119 had a clinical severe infection, six with a critical illness, and one with severe pneumonia (infant aged 0–6 days with fast breathing) and 24 cases had pneumonia (infant aged 7–59 days with only fast breathing) (Table 4 and Fig 3).

### Adherence to simplified antibiotic treatment for PSBI cases at PHCs and CHCs

Only 46% (58 out of 126 CSI, CI and severe pneumonia cases) completed 7-days of injection gentamicin and a third did not receive any injection. However, the majority of pneumonia (fast breathers 7–59 day old) cases (79.2%; 19/24) received all 14 doses of amoxicillin (Table 6).

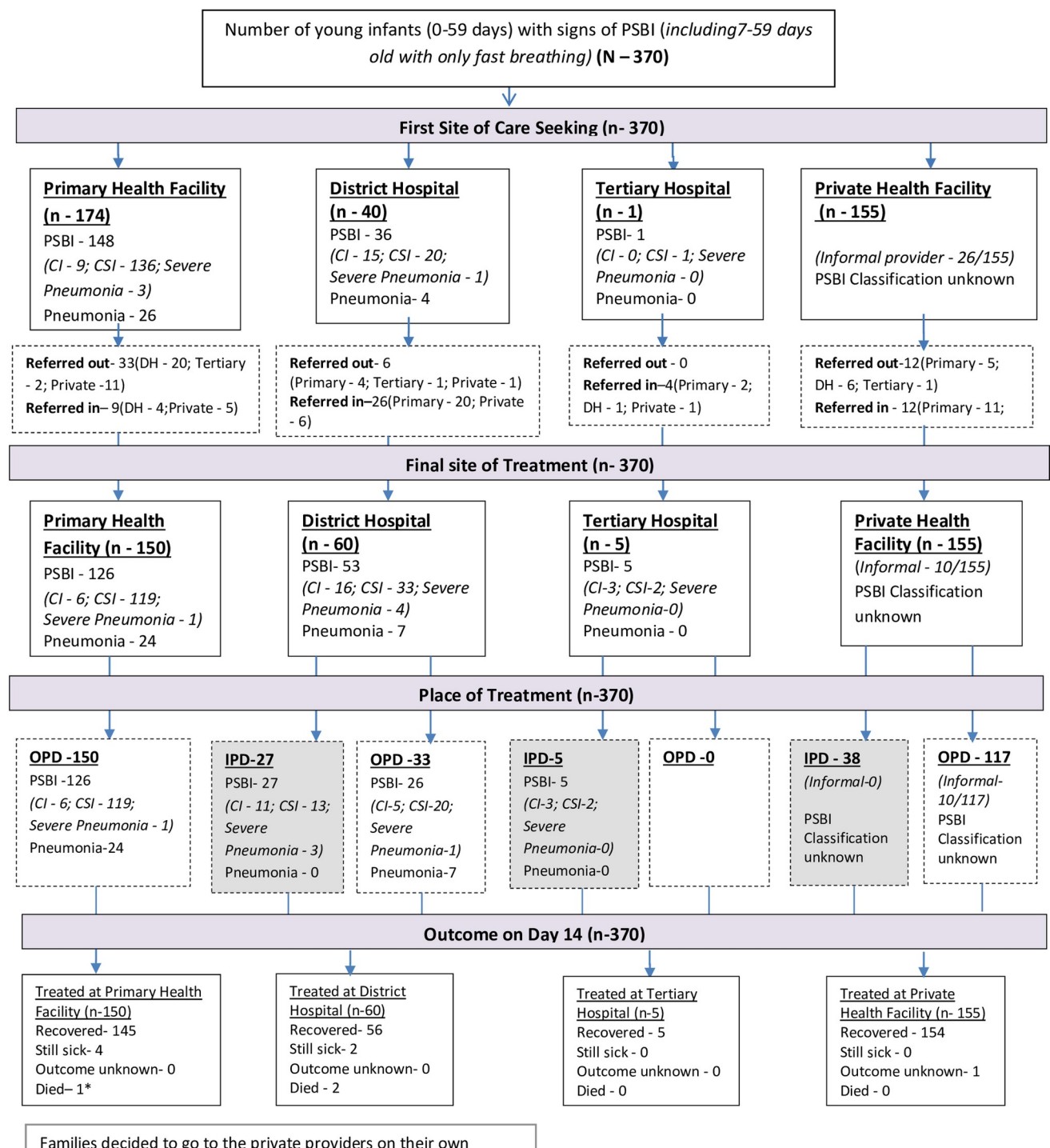

Fig 3. Flow diagram of identification, place of management and outcome of young infants with signs of Possible Serious Bacterial Infection (PSBI) (Aug 2017 –Mar 2019).

**Table 4. Identification, the first point of care, place of treatment and number of deaths of in young infants with PSBI signs across six study quarters (N- 370).**

| Implementation Quarters | Quarter1 | Quarter2 | Quarter3 | Quarter4 | Quarter5 | Quarter6 | Total |
|---|---|---|---|---|---|---|---|
| Time period | Aug-2017 to Oct-2017 | Nov-2017 to Jan-2018 | Feb-2018 to Apr-2018 | May-2018 to July-2018 | Aug-2018 to Oct-2018 | Nov-2018 to Jan-2019 | Aug-2017 to Jan-2019 (Total 6 quarters) |
| 2.1. Total number of live births occurred | 259 | 740 | 1,002 | 1,253 | 1,244 | 772 | 5,270 |
| 2.2. PSBI infant identified | 38(14.7) | 10(1.4) | 49(4.9) | 117(9.3) | 108(8.7) | 48(6.2) | 370(7.0) |
| 2.3. PSBI infant identified by | | | | | | | |
| • ASHA | 7(18.4) | 2(20.0) | 5(10.2) | 10(8.6) | 13(12.0) | 5(10.4) | 42(11.4) |
| • Families | 29(76.3) | 8(80.0) | 44(89.8) | 106(90.6) | 95(88.0) | 42(87.5) | 324(87.6) |
| • ANM | 1(2.6) | 0(0.0) | 0(0.0) | 0(0.0) | 0(0.0) | 0(0.0) | 1(0.3) |
| • Doctors | 1(2.6) | 0(0.0) | 0(0.0) | 1(0.8) | 0(0.0) | 1(2.1) | 3(0.8) |
| 2.4. The first point of care* | | | | | | | |
| • Government primary facility (PHC/CHC) † | 12(31.6) | 3(30.0) | 23(46.9) | 51(43.6) | 61(56.5) | 24(50.0) | 174(47.0) |
| • District Hospital | 4(10.5) | 2(20.0) | 5(10.2) | 13(11.1) | 13(12.0) | 3(6.3) | 40(10.8) |
| • Tertiary | 0(0.0) | 0(0.0) | 0(0.0) | 0(0.0) | 1(0.9) | 0(0.0) | 1(0.3) |
| • Private facility | 22(57.9) | 5(50.0) | 21(42.9) | 53(45.3) | 33(30.6) | 21(43.8) | 155(41.9) |
| 2.5. Place of treatment* ‡ | | | | | | | |
| • Government primary facility (PHC/CHC) † | 11(28.9) | 3(30.0) | 21(42.9) | 46(39.3) | 47(43.5) | 22(45.8) | 150(40.5) |
| • District Hospital | 6(15.8) | 3(30.0) | 7(14.3) | 19(16.2) | 20(18.5) | 5(10.4) | 60(16.2) |
| • Tertiary | 0(0.0) | 0(0.0) | 2(4.1) | 1(0.8) | 2(1.8) | 0(0.0) | 5(1.4) |
| • Private facility | 21(55.3) | 4(40.0) | 19(38.8) | 51(43.6) | 39(36.1) | 21(43.8) | 155(41.9) |
| 2.6 No. of Deaths identified§¶ | 1(2.6) | 0(0.0) | 0(0.0) | 0(0.0) | 1(0.9) | 1(2.1) | 3(0.8) |

* None of the case represented and managed in Sub Centres

†Government primary health care facility (PHC/CHC) includes: primary health center (PHC) and community health center (CHC)

‡Numbers mentioned in the *Place of treatment* are different from the *First point of care* as patients were referred from first point of care to final place of treatment (see Fig 3 for further details)

§Denominator is number of PSBI infant identified in the respective quarter

¶none of the FB only (7–59 days old) case died.

## Follow up and outcome of PSBI treated in PHCs/CHCs

Out of 150 PSBI cases who received simplified outpatient treatment for PSBI at PHC/CHC, 117 infants (79%) came for WHO-recommended mandatory follow up on Day-4 at the health facility (Table 6). Four infants (2.7%) had clinical treatment failure: two had stopped simplified antibiotic treatment and two did not receive injection gentamicin at all. One infant, who received complete 7 days of simplified treatment and recovered, died suddenly at home on the 15th day after being enrolled.

## Treatment outcome of the PSBI case treated in other facilities

Out of 65 PSBI cases managed at the district or a tertiary care hospital, 93.8% improved (61/65). Two with signs of critical illness died in the district hospital (Table 6). Two infants with pneumonia (7–59 days) did not accept treatment at PHC/CHC and received 14 doses of oral amoxicillin treatment from pediatric OPD of the district hospital, also recovered. All PSBI patients managed in private health facilities recovered, except one whose outcome is unknown (Fig 3).

**Table 5. Post-natal home visitation by ASHAs during first four quarters (Aug 2017—Jul 2018) of study period*.**

| Post-natal home visits by ASHAs | Based on the official record submitted by ASHA (N– 1475) † | Based on the call from the research team to families (*complete information available about the scheduled visits*) (N– 2001) ‡ |
|---|---|---|
| | n (%) | n (%) |
| • Day 1 | 366(24.8) | 372(18.6) |
| • Day 3 | 1341(90.9) | 1079(53.9) |
| • Day 7 | 1414(95.9) | 923(46.1) |
| • Day 14 | 1385(93.9) | 880(44.0) |
| • Day 21 | 1324(89.8) | 799(40.0) |
| • Day 28 | 1250(84.7) | 785(39.2) |
| • Day 42 | 1168(79.2) | 854(42.7) |
| • 0 visits | 0 (0%) | 336(16.8) |

*Total no. of live births during Aug 2017-Jul 2018–3,254

†Refusal of ASHA to share records of 1,779 mother–child dyads

‡Total no. of calls made to the families—16,977

## Sickness other than PSBI identified during the IR

Out of 5270 live births, 296 (5.6%) young infants had an illness other than PSBI. Out of 296, 126 (42.6%) had a local infection, 109 (36.8%) had diarrhoea, 47 (15.9%) had jaundice and in 13 (4%) infants with low weight had feeding difficulties but without any sign of PSBI (S2 Table).

## Discussion

Our coverage of identifying and treating PSBI cases was 70% (370 out of 5270 live births) when compared to an average 10% PSBI prevalence in young infants [2–5]. Among those who came to public health primary care facilities (PHCs/CHCs), less than 20% accepted a referral for hospitalization; remaining were managed in outpatients with simplified antibiotics; only one died (1/370; 0.3%) and rest recovered. Two-fifth of the PSBI infants were managed at public health facilities including district and tertiary care hospitals. Adherence to simplified antibiotic therapy for PSBI infants in the PHCs and CHCs gradually improved during the life of the project. Cases of the 7–59 day old fast breathers, received oral amoxicillin and recovered uneventfully. By the end of the implementation research, the majority of the families were able to recognize sick young infants and sought timely care.

In this implementation research, the PHCs and CHC became functional to manage the sick young infants with PSBI. The related supplies like antibiotics were made available with no stock-outs. PHC and CHC doctors, who were initially not confident to handle the infants less than 6 months, started managing PSBI with structured hand-holding and mentoring support. Lack of confidence amongst PHC and CHC doctors to assess and manage (including giving injections) to sick newborns and young infants has been reported in earlier studies from other parts of India [38–41]. The primary care physicians were particularly apprehensive about administering gentamicin injections in the first two quarters but the adherence gradually improved during the project. Variable adherence for treatment and follow up in the primary care including deviation from pre-referral protocol perhaps can partly be attributed to prescribers' conviction, confidence and caregivers' perception of the severity of the illness, perception on the recovery of the child, and limited counseling and communication between doctors and the mothers [19, 42–44].

**Table 6. Detail of treatment and follow-up of young infants with PSBI signs according to their clinical sub-categories (N = 370).**

| Parameters | Total PSBI cases | Critical illness (CI) | Clinical severe infection (CSI)* | Fast breathing only (7–59 days) |
|---|---|---|---|---|
| **A. Case Management and outcome of the PSBI cases who were treated at a government primary health facility (PHC and CHC)** | | | | |
| **4.1. Number of PSBI cases treated** | **n = 150** | **n = 6** | **n = 120** | **n = 24** |
| **4.2. No of PSBI cases who completed recommended simplified antibiotic treatment** | 68(45.3) | 2(33.3) | 47(39.2) | 19(79.2) |
| **4.3. Doses of antibiotic received—Gentamicin** | | | | |
| • 7 injections | 58(38.7) | 3(50.0) | 55(45.8) | - |
| • ≥2<7 injections | 24(16.0) | 2(33.3) | 21(17.5) | 1(4.2) † |
| • 1 injection | 6(4.0) | 1(16.7) | 5(4.2) | - |
| • 0 injection | 62(41.3) | 0(0.0) | 39(32.5) | 23 (96) |
| **4.4 Doses of antibiotic received—Amoxicillin** | | | | |
| • 14 doses | 102(68.0) | 4(66.7) | 79(65.8) | 19(79.2) |
| • 7–13 doses | 22(14.7) | 0(0.0) | 17(14.2) | 5(20.8) |
| • ≥1< 7 doses | 4(2.7) | 1(16.7) | 3(2.5) | 0(0.0) |
| • 0 dose of amoxicillin | 22(14.7) | 1(16.7) | 21(17.5) | 0(0.0) |
| **4.5. Follow-up visits of PSBI cases during treatment (simplified antibiotic regimen)** | | | | |
| • Follow-up on Day 4 | 117(78.0) | 6(100) | 93(77.5) | 18(75.0) |
| • Completed all follow-up visits (Day 4 and Day 7) | 92(61.3) | 4(66.7) | 77(64.2) | 11(45.8) |
| • Partial followed-up (Either day 4 or Day 7) | 26(17.3) | 2(33.3) | 17(14.2) | 7(29.2) |
| • No follow-up | 32(21.3) | 0(0.0) | 26(21.7) | 6(25.0) |
| **4.6. Treatment outcomes for PSBI cases (simplified antibiotic regimen)** | | | | |
| • Declared as 'Clinical treatment success' | 145(96.7) | 6(100.0) | 116(96.7) | 23(95.8) |
| • Declared as 'Clinical treatment failure'‡ | 4(2.7) | 0(0.0) | 3(2.5) | 1(4.2) |
| • Death | 1(0.7) | 0(0.0) | 1(0.8) | 0(0.0) |
| **B. Treatment outcome of PSBI cases treated at the District hospital/Tertiary Hospital (based on follow up on day 14 after diagnosis of the illness)** | **n = 65** | **n = 19** | **n = 39** | **n = 7** |
| • Recovered | 61(93.8) | 16(84.2) | 38(97.4) | 7(100) |
| • Still sick | 2(3.1) | 1(5.3) | 1(2.6) | - |
| • Outcome unknown | - | - | - | - |
| • Died | 2(3.1) | 2(10.5) | - | - |
| **C. Treatment outcome of PSBI cases treated by a private provider (based on follow up on day 14 after diagnosis of the illness)    §** | **n = 155** | | | |
| • Recovered | 154(99.4) | - | - | - |
| • Still sick | - | - | - | - |
| • Outcome unknown | 1(0.6) | - | - | - |
| • Died | - | - | - | - |

*Includes one infant 0–6 days old with fast breathing (severe pneumonia)

†One fast breathing only (7–59 days) case received injection gentamicin

‡ *Definition of Treatment Failure*: the appearance of any new sign of CI or CSI up to day 8 of treatment (worsening) **or** persistence of all presenting signs of CSI or CI on day 4 **or** persistence of any presenting sign by day 8 of treatment

§Case classification was not known for cases managed at a private facility

Similar to the strategy adopted by the current study, technical support to manage PSBI through a linkage between the PHCs and their feeder hospital pediatricians and/or neonatologists provided mentorship and encouraged physicians in the primary care setting to manage PSBI in infants in Nigeria [45]. Competency-based pre and in-service training complemented

by supportive supervision instilled confidence along with skill improvement amongst primary care physicians and mid-wives to perform newborn resuscitation in actual field conditions in Afghanistan [42], and South Africa [46] and manage sick young infants with PSBI more recently in India [39–41].

ANMs treated only two PSBI patients during the study at PHC/CHC under the doctor's supervision. GOI guideline recommends ANMs to manage PSBI cases at sub-centers, but despite the availability of supplies and training these could not be made functional. Formative research indicated that ANMs did not consider themselves as treatment providers and were afraid of administering therapeutic injections. The sub-centers where ANMs conducted their routine ANC and immunization clinics, suffered from poor infrastructure, non-availability of ANMs most of the time for medical consultation (she is travelling to villages under her charge for different program activities) and perception of the community about sub-centers not as places for management of severe ailments like PSBI. The majority of ANMs were not aware of the government permission to ANMs for assessment and management of PSBI infants including administration of injection gentamicin. Most of them perceived that assessment and management of sick young infants was the primary responsibility of the medical officers and thereafter if the doctors assigned them any responsibility, they could comply with it. We noted that ANMs developed confidence in administering injection gentamicin to PSBI after they observed the PHC/CHC doctors prescribing and administering these injections to sick young infants. A recent qualitative study from Pune on the performance of ANMs and three implementation research studies done in Maharashtra, Uttar Pradesh and Himachal Pradesh in India demonstrated the hesitation of ANMs to prescribing medicine for the same reasons [39–41, 47].

According to the HBNC program, home visitation by the ASHAs is key to empower mothers and also identify and prompt mobilization of sick neonates for appropriate care-seeking and management [22]. In the present study, ASHAs had identified little over one-tenth of all PSBI cases in the study area compared to none during the six months before the launch of the IR in the area. The poor performance of ASHAs for home visitation, and documentation despite efforts by the district health authorities and three rounds of skill-building exercises by the research team, was of concern; hence post-natal home visitations could not be leveraged sufficiently for the identification of sick infants. ASHAs were expected to prepare a line list that contained the names of the beneficiaries due for immunization, which was closely monitored, supervised and accounted for and hence this activity were performed with consistency and quality. On the other hand, there was an almost complete absence of on-ground supervision of quality and quantity of ASHAs' post-natal home visitation and scrutiny of the HBNC forms filled by them. The investigators felt that this was the major reason for poor performance despite incentives and skill-building workshops. Mothers had reported poor performance of ASHAs (<25%) for advice/counseling regarding obstetric danger sign assessment and neonatal care in Karnataka [48]. Issues like involvement in multiple programs, delayed incentive payments and lack of coordination with ANMs-were some of the other factors identified in the recently published IR from three different parts of the country [39–41].

We adopted a multi-channel targeted community mobilization approach to empower the families and mothers, which enabled them to identify 88% of the sick young infants and seek care in a timely manner. Overall only one PSBI patients died out of sick young infants managed on an outpatient basis at the PHCs/CHCs during the study period. In the study from Malawi, 85% of PSBI cases were identified by families and out of 378 PSBI cases, only one CSI baby died [49]. In comparison, the case fatality rate in a similar study from the Lucknow site in India was much higher, where families of many PSBI cases refused to accept referral or treatment at a public health facility and numerous families faced delays in getting appropriate

treatment at a government hospital [40]. Low PSBI mortality in the current observational study could therefore be attributed to early identification of illness and timely care seeking by the mothers and families and prompt treatment at the health facilities. Several innovative demand-side strategies to educate mothers and families in essential newborn care, identification of sickness in young infants, timely and appropriate care-seeking have been assessed through several research studies [50–56], but the results have been variable, i.e., 4% to 30% improvement in care-seeking [53] and up to 52% reduction in neonatal mortality [52, 56].

In a recent study from rural Bangladesh, care-seeking remained high with private providers (95%), predominantly village health doctors (over 80%) [57]. Caregivers in several low and low-middle income countries indicate distrust in government hospital doctors, inconsistent availability of health personnel and medicines as some of the important reasons for choosing private sector providers. However, when families perceived sickness to be severe enough, they sought higher-level care, from either the public or private sector [19]. In our study, although there was a limitation about the quality of data about PSBI infants treated at private facilities, 42% sought care from them. The observation was consistent with data from implementation research on integrated community case management in the same study area by another research group [30] and in Maharashtra [41] where private providers constitute an important service provider and need program attention.

The TSU was constituted as an interface between the program and the technical support for the implementation research to ensure smooth implementation and to oversee the developments on the ground. On the advice of the TSU, the research team co-implemented the PSBI program with health personnel during the first six months for better insight into implementation challenges, which helped to better contextualize strategies. It thus played an important role in the implementation of this intervention in the program setting. The establishment of such an institutional platform at the district level with technical support from local medical college and or public health institution can play a facilitator role when new interventions are rolled out or implementation challenges arise within the existing programs [39–41].

The results from our observational feasibility study need to be seen in light of its limitations. We were able to identify a much smaller number of infants with fast breathing due to a concurrent study about the management of the fast breathers by ASHAs, which led to relatively lower coverage of the sick young infants. But it is unlikely to influence the documentation of our experience of using simplified antibiotic including injectable antibiotics for other more severe categories of PSBI. Home visits by ASHAs were irregular and of poor quality for empowering the families and mothers, which led to a relatively small proportion of sick young infants by them. ASHAs are permanent member of the public health field team and can provide the sustainability to PSBI program, so the district authorities need to follow-up to increase the postnatal home visits. We did not cross-check the antibiotic doses administrated by the public facility doctors. Thus we could only comment on the effectiveness of simplified antibiotic therapy but cannot comment on the drug dosage and its response. Finally, for various operational and administrative reasons, we had to terminate the study without executing the panned exit strategy prepared in consultation with the TSU, which is likely to influence the sustainability of program refinements after the exit of the research team.

## Conclusions

In conclusion, we identified some key implementation barriers such as irregular and poor quality home visitation by the ASHAs, poor ability of the families and mothers to identify danger signs in their young infants, lack of confidence of primary care physicians at PHCS/CHCs to manage sick young infants and reluctance of the ANMs to consider themselves as care

providers. Most of the bottlenecks could be resolved by the district and state authorities and technical assistance from experts through leveraging existing resources and developing contextualized strategies. We demonstrated the feasibility of implementing management of PSBI on an outpatient basis when referral to a hospital was not feasible in a program setting by focusing on hand holding and confidence building of the primary care physicians, making the CHCs and PHCs functional and targeted social mobilization to empower the mothers and families for early recognition of sickness in their young infants.

## Supporting information

**S1 Checklist. Observation checklist for medical officers.**
(DOCX)

**S1 Fig. Identification, first point of care, place of treatment and number of deaths of in young infants with PSBI signs across six study quarters (N- 370).**
(DOCX)

**S1 Table. List of private providers who managed 155 PSBI infants during implementation research at district Palwal (Haryana, India).**
(DOCX)

**S2 Table. Total sickness identified in 0–59 days old infants during implementation period (Aug 2017—Jan 2019).**
(DOCX)

**S1 Data. INCLEN Palwal PSBI IR crf1 (n-5270).**
(XLSX)

**S2 Data. INCLEN Palwal PSBI IR crf2 (n-5270).**
(XLSX)

**S3 Data. INCLEN Palwal PSBI IR crf3 (n-370).**
(XLSX)

**S4 Data. INCLEN Palwal PSBI IR crf4 (n-370).**
(XLSX)

**S5 Data. INCLEN Palwal PSBI IR crf5 (n-370).**
(XLSX)

## Acknowledgments

We thank the Department of Health, Government of Haryana for facilitating the implementation research. We extend our gratitude to the public sector health personnel including doctors, ANMs, and ASHAs engaged in service delivery, community leaders of the area, district education department and school principals for supporting community mobilization efforts.

## Author Contributions

**Conceptualization:** Rupak Mukhopadhyay, Narendra Kumar Arora, Rakesh Gupta, Samira Aboubaker, Rajiv Bahl, Shamim Ahmad Qazi.

**Data curation:** Rupak Mukhopadhyay, Narendra Kumar Arora.

**Formal analysis:** Rupak Mukhopadhyay, Narendra Kumar Arora, Priya Limbu, Ramesh Poluru.

**Funding acquisition:** Narendra Kumar Arora.

**Investigation:** Rupak Mukhopadhyay, Narendra Kumar Arora.

**Methodology:** Rupak Mukhopadhyay, Narendra Kumar Arora, Suresh Dalpath, Ajay Khera, P. K. Prabhakar, Harish Chellani, Kailash Chander Aggarwal, Ratan Gupta, Sugandha Arya, Samira Aboubaker, Rajiv Bahl, Shamim Ahmad Qazi.

**Project administration:** Rupak Mukhopadhyay, Narendra Kumar Arora, Pradeep Kumar Sharma, Geetanjali Kataria, Rakesh Kumar Singh, Yogesh Malik, Saket Kumar, Yasir Bin Nisar, Shamim Ahmad Qazi.

**Resources:** Narendra Kumar Arora.

**Supervision:** Rupak Mukhopadhyay, Narendra Kumar Arora.

**Validation:** Rupak Mukhopadhyay, Narendra Kumar Arora.

**Writing – original draft:** Rupak Mukhopadhyay, Narendra Kumar Arora.

**Writing – review & editing:** Yasir Bin Nisar, Shamim Ahmad Qazi.

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
