## [Decision Letter · Decision Letter 0]

1 Sep 2020

PONE-D-20-18867

Lessons from an implementation research on community management of possible serious bacterial infection (PSBI) in young infants (0-59 days), when referral is not feasible in Palwal district of Haryana, India.

PLOS ONE

Dear Dr. Arora,

Thank you for submitting your manuscript to PLOS ONE. After careful consideration, we feel that it has merit but does not fully meet PLOS ONE’s publication criteria as it currently stands. Therefore, we invite you to submit a revised version of the manuscript that addresses the points raised during the review process.

There are extensive suggestions that must be faced to correct defects and make the manuscript to improve. Please report each change with the correction you´ve done in a document. An English revision is also necessary.

We look forward to receiving your revised manuscript.

Kind regards,

Ricardo Q. Gurgel, PhD

Academic Editor

PLOS ONE

Journal Requirements:

Reviewers' comments:

Reviewer's Responses to Questions

**Comments to the Author**

1. Is the manuscript technically sound, and do the data support the conclusions?

Reviewer #1: Partly

Reviewer #2: Partly

2. Has the statistical analysis been performed appropriately and rigorously? 

Reviewer #1: No

Reviewer #2: No

3. Have the authors made all data underlying the findings in their manuscript fully available?

Reviewer #1: Yes

Reviewer #2: Yes

4. Is the manuscript presented in an intelligible fashion and written in standard English?

Reviewer #1: Yes

Reviewer #2: Yes

5. Review Comments to the Author

Reviewer #1: Reviewer’s comments

The Introduction is concise and focused.

Methodology

The authors describe some of the health indicators like Neonatal Mortality Rate and Infant Mortality Rate (lines 111-114). However it is not clear if these indicators are specific to the study area or are national rates. If these are specific to the study area then each indicator should be compared to the national rates.

This study to a large extent exposes the health care seeking behavior of the community. The authors should present some of the relevant health care seeking indicators of the study setting. In addition to mentioning the number of health facility deliveries they should provide the proportion of home deliveries or deliveries assisted by health personnel. Also, other health seeking behaviors like proportion of the population who self-medicate or use other community management (including use of traditional methods) should be stated.

Study settings: The authors give a general description of the health facility structure but do not give the exact staffing and other resources in the study district. They need to describe the staffing levels in the different levels of health care and other facilities for management of the newborn.

Study design: In Line 137 the authors have not described the implementation research model used in the study. This information is very important for implementation research. The authors should clearly state the implementation research principle and design that was used in the study. Did they apply any framework?

Line 154-155: This sentence may not be included. The authors should avoid such casual statements as it puts questions on the criteria for selection of the study team.

Formative research:

Lines 179-181: Some of the methods used in the formative research are not scientifically rigorous, e.g. the use of non-formal interaction with health care personnel would not give the most valid results. They could have used standard methods like Key Informant Interviews.

In lines 181-184: the authors do not state the data collection methods used in baseline assessments of some of the factors; e.g. Exploration of the attitude and confidence of health providers in managing sick young infants; Existing gaps in reporting and monitoring HBNC related indicators; Quality and frequency of home visitations, etc. Without this information it is difficult to verify the reliability of the data.

Lines 196-200: Were standardized observation checklists used?

The authors should specify which data collection methods or data collection tools were used to obtain the data. They should differentiate between the methods used for quantitative and qualitative data collection.

The data analysis section is not described in detail. This study has a large component of qualitative data. The authors should describe in detail how the data from the various qualitative methods were analyzed and represented. The analysis of the different quantitative data, similarly, should be clearly described.

Discussion

Lines 396-401: The authors should explore and clearly indicate the possible reasons for poor performance of the ANMs and ASHA since their study had extensive qualitative aspects. They need to discuss the limitations these personnel are facing e.g. what hinders them from being confident in treating the newborns; why would the ASHAs have incomplete data; could it be a lack of training? Is it poor motivation from working conditions? These issues need to come out clearly. The authors are in a better position to give recommendations on how to enhance the performance of these personnel.

Line 428-431: The mortality rate in this study is impressively low. This is very important data from this study, and the authors ought to magnify this information. The authors should discuss how the NMR in their study compares with the national rates and explain the reasons for the low mortality rate. These may form their stem for major recommendations.

432-442: The authors do not have much evidence to discuss the private health facilities. However they can expound on the effects their interventions had on the services in the public health facilities and the increased utilization.

In the section on limitations, the authors should discuss each limitation stating the effects it had on the findings in their study.

The authors should calculate and reflect the cost estimate for the implementation of the interventions in order to shed light on feasibility of sustainability of the program.

Conclusion section (461-469): The conclusions of the study are not well aligned to the objectives. The authors should review this section and align the objectives (lines 93-100) to the conclusion. The authors are advised to avoid the use of abbreviations in the conclusion, except for those globally recognized.

Writing style; linguistic expression

The authors should use italics only when necessary.

Lines 191-194: Improve the English sentence construction.

Lines 237-241: Improve sentence construction and grammar. Use appropriate punctuation marks.

Lines 253-254; Review the sentence.

Lines 443; improve on the sentence construction.

Reviewer #2: This an interesting and potentially very useful formative evaluation of an effort to implement guidelines for community management of PSBI in young infants as a treatment option when referral is not possible in the Palwal district of Haryana, India. The implementation effort was multi-phased and appears to have shown some significant and important results with respect to feasibility of implementation; the paper also highlights some important “lessons learned” from the experience which may be beneficial for scaling out similar types of interventions and preventing further infant mortality. The potential contributions of the paper in its current form, however, are undermined by (1) a lack of clarity in describing the implementation and evaluation processes as distinct processes (where possible) or overlapping entities (when applicable); (2) informed descriptions of the methods, with scientific (or pragmatic) justifications for decisions made; (3) lack of boundaries between description of methods and reporting of results; (4) presentation of results that is very difficult to follow and does not focus on key findings relating to the purview of the research, as an observational, pragmatic study of feasibility of implementing guidelines in these type of communities; and (5) a lack of recognition of the limitations of the study, particularly when it comes to questions around, e.g., establishing the “effectiveness” of a drug relative to questions about implementation feasibility. The following comments highlight changes recommended prior to this paper’s acceptance for publication.

Background

• This section is quite short & missing some key information—notably, while the authors note that referral for admission to hospitals in areas like the one under study are challenging, they don’t describe any of the barriers. This is crucial background for understanding why an alternative process like the guideline implementation being studied is a preferable alternative to simply addressing those barriers to better referral.

• The authors should also add in some additional detail about the PSBI problem—e..g., what are the current mortality rates in India from PSBI, what fraction of infant mortality could be prevented with better treatment for PSBI, and how much would mortality rates be expected to decline if GOI/WHO guidelines were followed?

• The authors also need to include information (somewhere—here if documented in prior research/reports) what the known bottlenecks/barriers to implementing these guidelines are (e.g., the cited “paucity of operational and contextual barriers”)—what were these barriers and how did they inform the current research project?

• The objectives as currently written are difficult to follow and do not clearly align with an implementation evaluation or understanding of current barriers—I would strongly encourage the authors to make these objectives more concrete and then use these objectives to organize the methods (both implementation and evaluation methods) and results reporting. As an example, for objective 1, it is unclear what the authors are referring to when they say they are “strengthening the existing health system for early identification”—does that mean they are assessing barriers, or adding new resources, or …? Similarly, with objective 2—what is meant by “prepare primary care facilities”? What sort of preparation or implementation support is being provided? and how does this differ with objective 3, where the PSBI program is “embedded”? Without clarity as to the concrete aims of these objective in terms of both the program implementation and evaluation, it is very difficult to evaluate whether or not the implementation effort and evaluation went according to plan or—more importantly—whether the feasibility identified would be translatable to other similar settings.

Methodology

• Study population: Some additional information on the “aspirational” aspect of the Palwal district would be appreciated to understand potential generalizability, namely how do the birth and mortality rates compare to those across India? Additionally, the authors never note or justify why this district was chosen for this work & how this might relate to the generalizability of the findings—for example, is this a case of “if it doesn’t work here, it likely won’t work in other districts” because barriers should be lower here?

• Health infrastructure: Would encourage structuring the table currently labeled as Panel 1 in a way that makes it easier to follow and compare across settings—for example, having columns indicating type of provider present, type of care provided, maybe the total population it serves and/or number present in the area studied (the latter of which was not information I saw in the current format). This presentation would help to highlight key differences, especially as they relate to big questions regarding access undergirding this project.

• Study design:

o Section should specify which implementation research principles were used and justify the selection

o Generally, this section, the steps/methods used & how they map onto the objectives was difficult to follow. Would strongly suggest including a diagram or figure that maps both the steps of the process and the pieces of the formative evaluation onto the overall study objectives.

o Details about the study protocols—especially for interviews, surveys, etc—are almost completely absent. Protocols for each of these data collection efforts should be (at least briefly) described, in a table or elsewhere. Also, this section consistently conflates research methods with results—for example, reporting final Ns for data collection instruments, rather than study protocols and procedures for recruiting individuals to complete measures, instruments, or interviews.

o Policy dialogue: Panel 2 is interesting and clearly very important, but no context is given for why certain departures from the WHO guidelines were determined and/or how they were justified. This is very important to understand as implementation efforts assume implementation of an evidence-based practice; if these guidelines are forcing departure from that evidence-base, it undermines the entire endeavor. Alternatively, tailoring that is done to address known barriers is an important part of the implementation process and thus provision of details about why these changes were made should be provided.

o Implementation phase: Specifics on the implementation strategy used should be provided ideally in line with Proctor, Powell & McMillen (2013) and/or as specified through the ERIC classification of implementation strategies (cf. Powell et al, 2015).

o Table 1: This table is clearly both very important and also incredibly onerous and almost entirely ineffective at communicating information. Additionally, the information included here seems to be related to results, not methods—so it should be moved, and the results should be presented in a more reader-accessible and organized way. One option may be to keep this table as supplemental information and highlight key results (in the results section!) that are curated by the authors as highlighting key lessons learned.

o Nudges: No rationale or justification is provided for using the nudge strategy—which is notable because evidence regarding the effectiveness of nudges in changing behaviors is mixed. Additionally, it is not clear where or at what stages nudges were developed, how they were integrated into the overall implementation strategy, what protocol for nudges were & whether these protocols were followed with fidelity, and/or what results they were expected to have. Finally, no citations are provided for definitions of or literature around behavioral nudges.

o Data collection/management: Again, results are presented here. No Ns should be presented in methods sections (other than population N or target Ns).

• Results

o As with the methods section, it is difficult to follow which results track from which stages of the process. I would again encourage picking a structure for presenting the overall study design (e.g., in phases), mapping the objectives/research questions to those phases, and then mapping results (in this section) to those phases/questions. As presented currently, the conflation of methods for evaluation and results of those evaluations are very convoluted and difficult to extricate to determine whether results are valid or reliable, or whether they answer questions of interest.

o Table 2: Unclear whether the days listed in Column 1 are ranges (e.g., is the second line reflecting visits that occurred between Day 1 and Day 3 or just on Day 3)? Also unclear why 0 visits is inapplicable to the partial information category. Finally, I didn’t see the response rate indicated anywhere in the table or text—is it correct that the response rate is (2001+487)/16,997 total calls, i.e., less than 15%?

o Table 3: The presentation of this table (as well as some of the text in the conclusion) suggests that there were some over-time improvements in implementation, however these effects are not apparent from this table—perhaps because there is so much information that is presented here? One suggestion would be to curate this table a bit more to focus on the results that were most interesting? For example, since sub-centres were never the first point of care nor place of treatment, these rows could be removed from the table & a footnote could indicate this absence of action. Additionally, if changes over time are of interest, proper statistical tests should be included. (Note that a multi-paneled figure reflecting trends would be a much more reader-friendly way to convey this information as well).

o PSBI management:

Unclear where the 10% live births with PSBI assumption comes from; this should be justified and sources referenced.

Table 4: Best practices in this table should be highlighted to indicate what proportion of cases were ‘high fidelity’. Additionally, having a separate category for 6 cases (Critical illness) is problematic—would combine with one of the other categories (or simply not include as its own column). I also don’t think sections B1 and B2 need to be in the table since they are not a focus of the implementation effort. Discuss in the text but exclude from the table.

Unclear what the purpose of is including section on sicknesses identified other than PSBI—this should be dropped, and other relevant information related to barriers to care should be expanded.

• Discussion:

o Unclear where the 70% treatment coverage rate came from—this should be defined precisely in the results section.

o Generally, conclusions need to be tempered for this paper, and the conclusions should remind the readers from the outset that this was an observational, feasibility study for implementation of a particular guideline. While this paper does seem to strongly support feasibility of this intervention, there are serious limitations related to the selection of the region, problems with data reporting, and of course the study design that limit generalizability of findings, and certainly limit any notions of effectiveness or causality of the implementation. This is particularly notable for the authors claim that this study “demonstrated that oral amoxicillin alone is effective in managing pneumonia cases”—such a claim simply cannot be made with a study like this as there is no control case; data is purely observational. Further, claims like this distract from the implementation focus of the study. The authors would be much better off focusing on the purview suggested by the title—lessons from implementation—rather than trying to make claims related to clinical effectiveness. Of course, this doesn’t mean that they cannot claim that low mortality rates and high treatment rates aren’t important; they are—but they are reflective of the potential for guidelines like this to be implemented in communities like this that have room for improvement; their “effect” on downstream clinical outcomes simply cannot be estimated without a more rigorous study design.

o As with the rest of the manuscript, the conclusion has several places where important references appear to be missing—e.g., for “recent implementation research” that describes adherence to antibiotics guidelines improving slowly over time.

o The limitations should reflect the limitations of the observational study design, potential threats to validity and generalizability related to the population of interest, and larger issues related to data quality and potential missing data. This section is very weak at the moment.

6. PLOS authors have the option to publish the peer review history of their article (what does this mean?). If published, this will include your full peer review and any attached files.

Reviewer #1: No

Reviewer #2: No

---

## [Author Response · Author response to Decision Letter 0]

15 Mar 2021

Reviewer #1: Reviewer’s comments

1. The Introduction is concise and focused.

• Response: Thank you. 

Methodology

2. The authors describe some of the health indicators like Neonatal Mortality Rate and Infant Mortality Rate (lines 111-114). However it is not clear if these indicators are specific to the study area or are national rates. If these are specific to the study area then each indicator should be compared to the national rates.

Response: We have revised the estimate. Now, study region NMR and IMR are presented as per Haryana State Maternal and Infant Death review System (MIDSR, 2015-16); State and National estimate is referred from NFHS 4 (National Family Health Survey 2015-16) . References have been updated accordingly. We have added the following sentence in the revised manuscript on page 7, line 151- 154-130. “Infant mortality rate in Palwal was 35/1000 live birth (Haryana-32.8 /1000; India-41/1000) and neonatal mortality rate was 21/1000 live birth (Haryana 22.1 /1000; India 30/1000).” 

3. This study to a large extent exposes the health care seeking behavior of the community. The authors should present some of the relevant health care seeking indicators of the study setting. In addition to mentioning the number of health facility deliveries they should provide the proportion of home deliveries or deliveries assisted by health personnel. Also, other health seeking behaviors like proportion of the population who self-medicate or use other community management (including use of traditional methods) should be stated.

Response: We have now revised the section that represents the care seeking behavior in study area. We have added information on home deliveries and deliveries conducted by skilled birth attendants (NFHS 4). Data on self-medication and careseeking behavior has also been added. We have added the following sentence in the revised manuscript on page 7, line 155-165.

“According to the HMIS (health management information system), [28] Palwal district recorded 24046 deliveries during the year 2018-2019; 77% (18615) were institutional and the remaining (23%; 5431) occurred in homes. Only 17% (946/5431) of the home deliveries were attended by any skilled birth attendant. Furthermore, a verbal autopsy study from the same area indicated that 25% of neonatal deaths were due to sepsis [12] . In 50% of death cases, the neonates were not taken to any health facility before death. The actual figure of newborns and young infants with PSBI who were not accessing a health facility might be even higher. The preferred choice of care-seeking for neonatal illness in the area has been unqualified village practitioners followed by government hospitals and private qualified practitioners [29]. In a recent study on community management of pneumonia from the same study area 78% of sick infants and children visited private care providers (qualified providers, chemist, traditional healers). [30]”

4. Study settings: The authors give a general description of the health facility structure but do not give the exact staffing and other resources in the study district. They need to describe the staffing levels in the different levels of health care and other facilities for the management of the newborn.

• Response: We have segregated the staffs in different levels of health care now in Panel 1 (Please see the fifth column) in the revised manuscript page 8-10 as given below: 

Panel 1: Government district health facility framework and description of facilities in the context of PSBI management

Sl. no Name of the Facility/ Institution Expected strength of health personnel Type of care provided Population covered Number of health facilities in the study area Number of health personnel in position during the study period for neonatal care

1 Accredited Social Health Activist ( ASHA) Sanctioned positions for the area-172

 Social mobilizer for promoting maternal and child health: ante-natal care; deliveries; immunization and sickness identification, counseling, 

door-to-door surveys for specific service recipients @One ASHA for approximately 1000 rural population Working from home

 172 

2 Sub-Centre

(SC) 1. Two ANMs 

2. one male multi-purpose health worker (MPW) Lower most health service delivery facility; provides basic antenatal care, immunization and treatment for minor illnesses 3000-5000 18 ANM: 29 

Male Multi Purpose Health worker -3 

3 Primary Health Centre (PHC) 1. Two qualified doctors (1-allopathic-MBBS and 1- Ayush Traditional medicine physician) 

2. One staff nurse 

3. One ANM

4. Other staff not directly involved with neonatal care: 1 pharmacist; 1 health educator and 2 health assistants PHC- first port of call to a qualified public sector doctor in rural areas for the sick and those who directly report or referred from Sub-Centres (5-6 SC fall under each PHC) for curative, preventive and promotive health care.

4-6 Bedded facility 20000-30000 3 PHC 1: Nagaljaat

Medical Officer: 0

Nurse: 1

ANM: 2

PHC 2: Kot

Medical Officer: 2

Nurse: 0

ANM:1

PHC 3: Uttawar 

Medical Officer: 2

Nurse: 0

ANM:1

4 Community Health Centre (CHC) 1. Five medical specialists (i.e., general surgeon, physician, gynecologist, anesthetist and pediatrician)

2. Four general duty officers (1 dental surgeon, 3 general medical officers 

3. Twenty one paramedical and support other staff. First referral unit for SCs and PHCs;

Facilities with one operation theater, X-ray machine, labour room and laboratory. 

Provides facilities for 24X7 obstetric care and specialist consultations

Beds: 20-30 80000-120000 3 CHC 1: Aurangabad

Specialist: 0

Medical Officers: 4

Nurse: 1

ANM: 3

CHC 2: Hodal

Specialist: 0

Medical Officers: 4

Nurse: 4

ANM: 4

CHC 3: Hathin 

Specialist: 0

Medical Officers: 5

Nurse: 4

ANM: 2

5 District Hospital (DH) Palwal district hospital is a 

200 bedded hospital.

The district hospital has 

1. Thirty seven specialists (including 3 paediatricians) 

2. One hundred and thirty two paramedical staffs ( including 90 Staff nurse) District Hospital is secondary referral level facility and provides comprehensive secondary health care services to the population in the district. District Hospital is expected to deliver Essential (Minimum Assured Services) and Desirable (which we should aspire to achieve) package of services. The services include OPD, indoor and emergency service. In addition, basic specialty services, newborn care, psychiatric services, physical medicine and rehabilitation services, accident and trauma services, dialysis services and anti-retroviral therapy. Average population of a district varies between 15 to 30 million 

Palwal district hospital covers a population of 

1 million 1 

Palwal Civil Hospital

Paediatrician:3

Medical officers:6

Nurse: 6

A deputy Chief Medical Officer [Deputy-CMO] supervised ANMs and ASHA coordinators [supervisors for ASHAs] 

Health administration in the district managed by Chief Medical Officer’s Office

6 Special Newborn Care Unit (SNCU) Managed by an adequately trained pediatrician, doctors, staff nurses and support staff to provide 24x7 services.

As per MOHFW, the expected staff strength for a SNCU should be

1. One doctor per 4 beds 

2. One nurse per 1.5 beds

( Palwal SNCU should have 5 doctors and 12 nurses) A SNCU is established at the district hospital and sub-district hospitals to provide care for sick newborns, i.e., all type of neonatal care except assisted ventilation and major surgeries. It is a separate unit close to the labour room with 12 or more beds.

Palwal DH- SNCU has 18 beds to manage sick newborns >3000 to 20000 annual delivery at district /sub-district level 1 Same as mentioned in row 5 under the district hospital

5. Study design: In Line 137 the authors have not described the implementation research model used in the study. This information is very important for implementation research. The authors should clearly state the implementation research principle and design that was used in the study. Did they apply any framework?

• Response: We have incorporated the study framework adopted from the RE-AIM framework. We have added the following sentence in the revised manuscript on page 10, line 184-185.

“RE-AIM framework [31] was adopted to evaluate the implementation of PSBI guidelines in the study (Fig 2).”

 Ref - Glasgow RE, Vogt TM, Boles SM. Evaluating the public health impact of health promotion interventions: the RE-AIM framework. Am J Public Health (1999) 89(9):1322–7. doi:10.2105/AJPH.89.9.1322

6. Line 154-155: This sentence may not be included. The authors should avoid such casual statements as it puts questions on the criteria for selection of the study team.

• Response: We considered policymakers and district level implementers as key stakeholders who were part of the policy and programme dialogue process. The Chief Medical Officer (Dr PK Sharma) has participated in the strategic planning, analysis and subsequent dissemination process; CMO travelled to Nairobi (Kenya) to make the presentation of the study findings to the international audience also; Deputy director made presentations in protocol finalization meeting in Lagos (Nigeria) while CMO made four national-level presentations. We hope that our explanation takes care of the reviewer’s concerns.

Formative research:

7. Lines 179-181: Some of the methods used in the formative research are not scientifically rigorous, e.g. the use of non-formal interaction with health care personnel would not give the most valid results. They could have used standard methods like Key Informant Interviews.

• Response: Non-formal interaction (NFI) or informal interaction is a scientifically valid qualitative method that is cited in the literature (please see references cited below). In our study, we asked questions that were related to their professional practices, skills and confidence in handling sick young infants. Thus the NFIs as a qualitative technique had better scope to explore and examine the domains more realistically in a real-world situation. All the NFIs had semi-structured interview guides. 

 References for NFIs:- 

o Chaturvedi, S., Ramji, S., Arora, N. K., Rewal, S., Dasgupta, R., Deshmukh, V., & for INCLEN Study Group (2016). Time-constrained mother and expanding market: an emerging model of under-nutrition in India. BMC public health, 16, 632. https://doi.org/10.1186/s12889-016-3189-4

o Gray DE. 2nd ed. Thousand Oaks, California: Sage Publications; 2009. Doing Research in the Real World

o Swain, J., & Spire, Z. (2020). The Role of Informal Conversations in Generating Data, and the Ethical and Methodological Issues. Forum Qualitative Sozialforschung / Forum: Qualitative Social Research, 21(1). doi:http://dx.doi.org/10.17169/fqs-21.1.3344

8. In lines 181-184: the authors do not state the data collection methods used in baseline assessments of some of the factors; e.g. Exploration of the attitude and confidence of health providers in managing sick young infants; Existing gaps in reporting and monitoring HBNC related indicators; Quality and frequency of home visitations, etc. Without this information, it is difficult to verify the reliability of the data.

• Response: We have revised the paragraph in light of the reviewer’s comments. Please refer line 259 to 278 in page 14 and 14

 “Formative research-part 1(June-August 2017): Baseline assessment included four components. i) Facility assessment: WHO facility scoring tool [33] was adapted to assess readiness for the availability of PSBI related supplies, manpower, service delivery and mechanisms of monitoring implementation of PSBI guideline (DH-1, CHC-3, PHC-3, sub-centers-18). ii) ASHAs’ perspectives about HBNC program, home visitation, sickness identification among young infants and care provision were obtained through knowledge attitude and practice (KAP) survey (n- 60; at the rate of at least one ASHA per village). iii) Perspectives of recently delivered (<6 months) and lactating mothers: In-depth interviews (n-30 mothers) and KAP survey with 150 mothers (at the rate of 3 randomly selected mothers from every village) to document mothers’ awareness of danger signs, care-seeking for their sick newborn and perceived value of home visitation by the ASHAs. iv) Towards the end of part 1 of the formative phase, both in-depth interviews and the quantitative survey indicated ground-level challenges about home visitation under HBNC by the ASHAs, assessment and management of young infants at the PHCs and CHCs and its influence on the care-seeking by the families for sick young infants. The respondents (surveys of mothers and ASHAs) had touched on these issues but in a restrained manner probably due to the sensitive nature of the problems. Non-formal interactions (NFI) [34, 35, 36] with primary care physicians working at CHCs and PHCs (n-5), pediatric specialists at district hospital (n-2), ASHAs (n-8), ANMs (n-5) and mothers (n-11) were undertaken to further explore these issues. Only senior investigators conducted NFIs which were considered critical to the success of the implementation research. No tape recording or notes were taken at the time of interaction but immediately after NFI, the summary of the interaction was noted down. “

9. Lines 196-200: Were standardized observation checklists used?

The authors should specify which data collection methods or data collection tools were used to obtain the data. They should differentiate between the methods used for quantitative and qualitative data collection.

• Response: 

a. A pregnancy survey was conducted using a standardized line listing formats consisting of some key information of pregnant women [name, age, last menstrual period (LMP), expected date of delivery (EDD), name of Husband, name of household head, date of pregnancy registration, number of antenatal care (ANC) visits]. 

b. Documentation of ASHA’s post-natal home visitation by calling of recently delivered mothers -was conducted using a call confirmation sheet. For every new delivery, a calling- scheduled for expected ASHAs visits were made. The call schedule was fixed at 48 hours after the expected AHSAs home visitation for the family. Research staff made a call to the families to confirm if home visits were conducted by ASHAs.

c. We did not carry out any real-time observation for ASHAs home visitation. We have removed the text referring to the above observation. 

d. Observation checklist of physician included - (1) if the physician asked how was the infant(2) asked for seven danger signs (3) examined the child (4) used IMNCI recording form. 

We have modified the write up ( line 279- 291 in page 15 and page 16 ) 

“Co-participatory formative research – part 2 (August 2017- January 2018): The purpose was to get first-hand insight into challenges and barriers to implementing PSBI guideline beyond that obtained under Part 1 of the formative phase. The co-participant implementation phase was limited to six villages (population-29000); served by 71 ASHAs, two SCs, one CHC and the district hospital. During this period the research team also undertook house to house survey to determine the pregnancies missed by the health system (ASHAs), tracked them for deliveries and called up families with newborn to verify the ASHAs’ scheduled visits (For every new delivery a schedule of expected home visits by the ASHA was made and research staff called up the families between 48 and 96 hours after the scheduled time to confirm the home visits.) At least one of the investigators observed the doctors assessing infants aged less than 59 days at the CHC (receiving patients from the above mentioned six villages) using a structured observation checklist ( Supporting Information S1.Checklist) . This component was part of a planned staged implantation scale-up of the study; after the end of the six months, the study was expanded to all 50 villages. “ 

10. The data analysis section is not described in detail. This study has a large component of qualitative data. The authors should describe in detail how the data from the various qualitative methods were analyzed and represented. The analysis of the different quantitative data, similarly, should be clearly described.

• Response: We have modified the data analysis section as per the reviewer’s feedback ( Line no – 358 to 387 in page no 18 and 19)

“ The research data collection team comprised of Indian traditional medicine graduates with master’s in public health, graduate social scientists and field workers with at least 10 years of schooling. The research personnel were trained as per their assigned tasks in a 3-5 days’ workshop. They collected data for during the formative research; medical graduates interacted with medical officers in primary health facilities in the first two quarters, conducted post-treatment follow up (on 8th and 14th day) of PSBI diagnosis irrespective of the place where they were treated and abstracted administrative data from official documents of ASHAs, ANMs, PHCs, CHCs and district hospital. The data included: pregnancy, live birth, home visitation by ASHAs, sick young infants assessed and managed at different public sector health facilities, PSBI case classification formats, drugs administered, place of management (outpatient or in-patient), adherence, and follow up on day four and seven after initiating the treatment from all government health facilities. 

As part of the community mobilization and engagement activities, the families were asked to inform the research team telephonically if they chose to take their sick young infants to private providers (Table S1). Data about obtaining treatment at private health facilities, clinical features at the time of seeking care and outcome was based on the recall by the mothers and families and available prescriptions. Information was obtained by the research team on the 8th and 14th day follow up post-initiation of treatment. The information about the case classification was not available for these cases. The processes and experiences of different aspects of implementation were recorded in the daily diaries of the study investigator and team leaders.

Quantitative data (KAP surveys of mothers and ASHAs, facility assessments, pregnancy and birth surveillance, assessment of sick young infants, treatment provided and outcome) was double entered in RedCap® double data module, for detecting inconsistencies and merged after validation. Data was exported to STATA® (Version 15.0) for analysis. Descriptive statistics were used to present coverage data, treatment adherence and cross-sectional survey data. 

Qualitative data: Transcripts of recorded interviews (IDIs conducted during formative research with health service personnel, mothers and ASHAs) were prepared and complemented with field notes taken during non-formal interactions. All transcripts were entered in IQDAS (INCLEN Qualitative Data Analysis Software) [37]. Data were free-listed and key axial and selective codes were generated for analysis.” 

Discussion

11. Lines 396-401: The authors should explore and indicate the possible reasons for the poor performance of the ANMs and ASHA since their study had extensive qualitative aspects. They need to discuss the limitations these personnel are facing e.g. what hinders them from being confident in treating the newborns; why would the ASHAs have incomplete data; could it be a lack of training? Is it poor motivation from working conditions? These issues need to come out clearly. The authors are in a better position to give recommendations on how to enhance the performance of these personnel.

• Response: We have elaborated the possible reasons now

ANM related issues ( Line -534 to 549 in page no 33 and 34)

 “Formative research indicated that ANMs did not consider themselves as treatment providers and were afraid of administering therapeutic injections. The sub-centers where ANMs conducted their routine ANC and immunization clinics, suffered from poor infrastructure, non-availability of ANMs most of the time for medical consultation (she is travelling to villages under her charge for different program activities) and perception of the community about sub-centers not as places for management of severe ailments like PSBI. The majority of ANMs were not aware of the government permission to ANMs for assessment and management of PSBI infants including administration of injection gentamicin. Most of them perceived that assessment and management of sick young infants was the primary responsibility of the medical officers and thereafter if the doctors assigned them any responsibility, they could comply with it. We noted that ANMs developed confidence in administering injection gentamicin to PSBI after they observed the PHC/CHC doctors prescribing and administering these injections to sick young infants. A recent qualitative study from Pune on the performance of ANMs and three implementation research studies done in Maharashtra, Uttar Pradesh and Himachal Pradesh in India demonstrated the hesitation of ANMs to prescribing medicine for the same reasons [47, 39, 40, 41].” 

 ASHA related issues ( Line -552 to 566 in page no 34 and 35)

“The poor performance of ASHAs for home visitation, and documentation despite efforts by the district health authorities and three rounds of skill-building exercises by the research team, was of concern; hence post-natal home visitations could not be leveraged sufficiently for the identification of sick infants. ASHAs were expected to prepare a line list that contained the names of the beneficiaries due for immunization, which was closely monitored, supervised and accounted for and hence this activity were performed with consistency and quality. On the other hand, there was an almost complete absence of on-ground supervision of quality and quantity of ASHAs’ post-natal home visitation and scrutiny of the HBNC forms filled by them. The investigators felt that this was the major reason for poor performance despite incentives and skill-building workshops. Mothers had reported poor performance of ASHAs (<25%) for advice/counseling regarding obstetric danger sign assessment and neonatal care in Karnataka [48]. Issues like involvement in multiple programs, delayed incentive payments and lack of coordination with ANMs-were some of the other factors identified in the recently published IR from three different parts of the country [39, 40, 41 ].”

12. Line 428-431: The mortality rate in this study is impressively low. This is very important data from this study, and the authors ought to magnify this information. The authors should discuss how the NMR in their study compares with the national rates and explain the reasons for the low mortality rate. These may form their stem for major recommendations.

• Response: We have incorporated some text on low mortality. However, considering the present study design and scope of the study we feel that we must not over-interpret the data. Please refer Line 574 – 580 in page no 35

“ Low PSBI mortality in the current observational study could therefore be attributed to early identification of illness and timely care seeking by the mothers and families and prompt treatment at the health facilities. Several innovative demand-side strategies to educate mothers and families in essential newborn care, identification of sickness in young infants, timely and appropriate care-seeking have been assessed through several research studies [50-56], but the results have been variable, i.e., 4% to 30% improvement in care-seeking [53] and up to 52% reduction in neonatal mortality[52, 56].” 

13. 432-442: The authors do not have much evidence to discuss private health facilities. However they can expound on the effects their interventions had on the services in the public health facilities and the increased utilization.

• Response: We did not document the newborn management practices of private health facilities and thus we could not describe the effect of our intervention in the private sector. As documented in our formative research and indicated in another study (Taneja et al), the communities in the study area preferred private health facilities to seek care for their sick young infants. 

Refer line 583 to 591 in page 35 and 36

Taneja S, Dalpath S, Bhandari N, Kaur J, Mazumder S, Chowdhury R, et al. Operationalising integrated community case management of childhood illnesses by community health workers in rural Haryana. ActaPaediatrica. 2018 Dec;107:80-8https://doi.org/10.1111/apa.14428 PMID: 30570793

14. In the section on limitations, the authors should discuss each limitation stating the effects it had on the findings in their study.

• Response: We have modified the section as suggested. 

Refer line no 601 to 614 in page 36

“The results from our observational feasibility study need to be seen in light of its limitations. We were able to identify a much smaller number of infants with fast breathing due to a concurrent study about the management of the fast breathers by ASHAs, which led to relatively lower coverage of the sick young infants. But it is unlikely to influence the documentation of our experience of using simplified antibiotic including injectable antibiotics for other more severe categories of PSBI. Home visits by ASHAs were irregular and of poor quality for empowering the families and mothers, which led to a relatively small proportion of sick young infants by them. ASHAs are permanent member of the public health field team and can provide the sustainability to PSBI program, so the district authorities need to follow-up to increase the postnatal home visits. We did not cross-check the antibiotic doses administrated by the public facility doctors. Thus we could only comment on the effectiveness of simplified antibiotic therapy but cannot comment on the drug dosage and its response. Finally, for various operational and administrative reasons, we had to terminate the study without executing the panned exit strategy prepared in consultation with the TSU, which is likely to influence the sustainability of program refinements after the exit of the research team.” 

15. The authors should calculate and reflect the cost estimate for the implementation of the interventions in order to shed light on feasibility of sustainability of the program. 

• Response: We are preparing a separate manuscript on cost estimate as it requires a fair amount of space due to separate methodology etc. It didn’t under the scope of the current manuscript so we didn’t include it. The other collaborating sites from India, which have already published similar implementation lessons didn’t include a cost estimate.

Goyal N, Rongsen-Chandola T, Sood M, Sinha B, Kumar A, et al. (2020) Management of possible serious bacterial infection in young infants closer to home when referral is not feasible: Lessons from implementation research in Himachal Pradesh, India. PLOS ONE 15(12): e0243724. https://doi.org/10.1371/journal.pone.0243724 PMID: 33351810

Awasthi S, Kesarwani N, Verma RK, Agarwal GG, Tewari LS, et al. (2020) Identification and management of young infants with possible serious bacterial infection where referral was not feasible in rural Lucknow district of Uttar Pradesh, India: An implementation research. PLOS ONE 15(6): e0234212. PMID: 32497092

Roy S, Patil R, Apte A, Thibe K, Dhongade A, et al. (2020) Feasibility of implementation of simplified management of young infants with possible serious bacterial infection when referral is not feasible in tribal areas of Pune district, Maharashtra, India. PLOS ONE 15(8): e0236355. https://doi.org/10.1371/journal.pone.0236355 PMID: 32833993

Conclusion section 

16. (461-469): The conclusions of the study are not well aligned to the objectives. The authors should review this section and align the objectives (lines 93-100) to the conclusion. The authors are advised to avoid the use of abbreviations in the conclusion, except for those globally recognized.

• Response: We have aligned the two. 

Please refer line 617 to 626 in page 37 

“In conclusion, we identified some key implementation barriers such as irregular and poor quality home visitation by the ASHAs, poor ability of the families and mothers to identify danger signs in their young infants, lack of confidence of primary care physicians at PHCS/CHCs to manage sick young infants and reluctance of the ANMs to consider themselves as care providers. Most of the bottlenecks could be resolved by the district and state authorities and technical assistance from experts through leveraging existing resources and developing contextualized strategies. We demonstrated the feasibility of implementing management of PSBI on an outpatient basis when referral to a hospital was not feasible in a program setting by focusing on hand holding and confidence building of the primary care physicians, making the CHCs and PHCs functional and targeted social mobilization to empower the mothers and families for early recognition of sickness in their young infants.”

Writing style; linguistic expression

17. The authors should use italics only when necessary.

• Response: Revised as per suggestion 

18. Lines 191-194: Improve the English sentence construction

• Response: Revised 

19. Lines 237-241: Improve sentence construction and grammar. Use appropriate punctuation marks.

• Response : corrected 

20. Lines 253-254; Review the sentence.

• Response: corrected

21. Lines 443; improve on the sentence construction.

• Response: corrected

Reviewer #2: 

This an interesting and potentially very useful formative evaluation of an effort to implement guidelines for community management of PSBI in young infants as a treatment option when referral is not possible in the Palwal district of Haryana, India. The implementation effort was multi-phased and appears to have shown some significant and important results with respect to feasibility of implementation; the paper also highlights some important “lessons learned” from the experience which may be beneficial for scaling out similar types of interventions and preventing further infant mortality. The potential contributions of the paper in its current form, however, are undermined by (1) a lack of clarity in describing the implementation and evaluation processes as distinct processes (where possible) or overlapping entities (when applicable); (2) informed descriptions of the methods, with scientific (or pragmatic) justifications for decisions made; (3) lack of boundaries between description of methods and reporting of results; (4) presentation of results that is very difficult to follow and does not focus on key findings relating to the purview of the research, as an observational, pragmatic study of feasibility of implementing guidelines in these type of communities; and (5) a lack of recognition of the limitations of the study, particularly when it comes to questions around, e.g., establishing the “effectiveness” of a drug relative to questions about implementation feasibility. The following comments highlight changes recommended prior to this paper’s acceptance for publication.

Background

22. This section is quite short & missing some key information—notably, while the authors note that referral for admission to hospitals in areas like the one under study are challenging, they don’t describe any of the barriers. This is crucial background for understanding why an alternative process like the guideline implementation being studied is a preferable alternative to simply addressing those barriers to better referral.

Response : Reasons added. 

Pleeasse refer line 101 to 107, in page 5 

“In the study in Delhi slums [8], only 24% of the PSBI infants complied with hospital referral; the reasons for non-compliance were – child not perceived to be ill enough for hospitalization by the family, no one to accompany the mother or care for other siblings, waiting for the response to medicines advised mostly by the unqualified local practitioners, sought medicine from other physicians, unpleasant past experiences of the hospital, and trial of home remedies. In the Bangladesh study [13], 28.5% of PSBI infants died when left untreated or treated by unqualified health providers. “

23. The authors should also add in some additional detail about the PSBI problem—e..g., what are the current mortality rates in India from PSBI, what fraction of infant mortality could be prevented with better treatment for PSBI, and how much would mortality rates be expected to decline if GOI/WHO guidelines were followed?

• Response: Statistics added at the beginning of the introduction. 

Please refer line 94 to 101 in page 5 

Over half a million neonates died in 2019 in India and 33% of these deaths were due to one or more infectious causes.[1]. Around 10% -13% of newborns and infants below two months develop symptoms and signs suggestive of possible serious bacterial infection (PSBI) [2,3,4,5]. Although the recommended treatment for possible serious bacterial infection (PSBI) is hospitalization [6,7 ], referral to higher facilities and hospital admission remains challenging in several low and middle-income countries [8,9, 10, 11 ]. A verbal autopsy study from the study area (then called Mewat district) showed that 22.3% of neonatal deaths were due to sepsis and infections; and for 52.6% of these neonates, the families did not seek care outside their homes before death. [12]

24. The authors also need to include information (somewhere—here if documented in prior research/reports) what the known bottlenecks/barriers to implementing these guidelines are (e.g., the cited “paucity of operational and contextual barriers”)—what were these barriers and how did they inform the current research project?

• Response: We have now mentioned some of the known barriers from other country settings. Please note that, we have already mentioned in the introduction that there was a “paucity of evidence on contextual and operational barriers as far as India is concerned ”- Our study ( and other three sites from India) are expected to shed some light on the understanding of bottlenecks as these studies have captured the implementation process on a real-time basis. The implementation of the guideline truly happened for the first time in four sites in India, thus we could not refer to other data that highlighted the barriers of the implementation. 

However, we have mentioned what are the challenges in care seeking for sick young infants. Please refer line -116 to 118 in page 5 and 6 

“In a resource-poor setting care-seeking for young infants remains challenging due to distance to hospital, accessibility, affordability, time cost, wage loss, concern about the quality of care or attitude of the health workers, and cultural issues. [8,9,10,11, 18, 19 ] ”

25. The objectives as currently written are difficult to follow and do not align with an implementation evaluation or understanding of current barriers—I would strongly encourage the authors to make these objectives more concrete and then use these objectives to organize the methods (both implementation and evaluation methods) and results reporting. As an example, for objective 1, it is unclear what the authors are referring to when they say they are “strengthening the existing health system for early identification”—does that mean they are assessing barriers, or adding new resources, or …? Similarly, with objective 2—what is meant by “prepare primary care facilities”? What sort of preparation or implementation support is being provided? and how does this differ with objective 3, where the PSBI program is “embedded”? Without clarity as to the concrete aims of these objective in terms of both the program implementation and evaluation, it is very difficult to evaluate whether or not the implementation effort and evaluation went according to plan or—more importantly—whether the feasibility identified would be translatable to other similar settings.

• Response : We have modified the language to make it clearer.

Please refer line 121- 124 in page no 6 

“The objectives of this implementation research were to understand the programmatic bottlenecks, determine the feasibility and acceptability of contextually modified implementation strategies and increase access to PSBI treatment using WHO PSBI management guideline when the referral is not feasible [15]”

Methodology

26. Study population: Some additional information on the “aspirational” aspect of the Palwal district would be appreciated to understand potential generalizability, namely how do the birth and mortality rates compare to those across India? Additionally, the authors never note or justify why this district was chosen for this work & how this might relate to the generalizability of the findings—for example, is this a case of “if it doesn’t work here, it likely won’t work in other districts” because barriers should be lower here?

• Response: Birth and mortality rate have been added. Also, the rationale for choosing the district has been given

Please refer line 146- 165 in page 7. 

“Developmental and Environmental Surveillance Site) (www.somaarth.org), comprising of a population of 199,143 in 50 villages of three administrative blocks (Hathin, Hodal and, Palwal) (Fig 1). In the Haryana Vision 2030 document, [24] Palwal has been ranked lowest in the human development index and fares poorly across all three indicators of human development, health, education and per capita income. Palwal district is one of the aspirational districts of the Government of India (i.e. with amongst lowest development indicators). The crude birth rate of the district was 26/1000 (compared to 20.3 in Haryana and 20.4 in National) in 2017. [25] The infant mortality rate in Palwal was 35/1000 live birth (Haryana-32.8 /1000; India-41/1000) and the neonatal mortality rate was 21/1000 live birth (Haryana 22.1 /1000; India 30/1000). [26 27] According to the HMIS (health management information system), [28] Palwal district recorded 24046 deliveries during the year 2018-2019; 77% (18615) were institutional and the remaining (23%; 5431) occurred in homes. Only 17% (946/5431) of the home deliveries were attended by any skilled birth attendant. Furthermore, a verbal autopsy study from the same area indicated that 25% of neonatal deaths were due to sepsis [12] . In 50% of death cases, the neonates were not taken to any health facility before death. The actual figure of newborns and young infants with PSBI who were not accessing a health facility might be even higher. The preferred choice of care-seeking for neonatal illness in the area has been unqualified village practitioners followed by government hospitals and private qualified practitioners [29]. In a recent study on community management of pneumonia from the same study area 78% of sick infants and children visited private care providers (qualified providers, chemist, traditional healers). [30]”

27. Health infrastructure: Would encourage structuring the table currently labeled as Panel 1 in a way that makes it easier to follow and compare across settings—for example, having columns indicating type of provider present, type of care provided, maybe the total population it serves and/or number present in the area studied (the latter of which was not information I saw in the current format). This presentation would help to highlight key differences, especially as they relate to big questions regarding access undergirding this project.

• Response: Table has been modified as suggested.

Please refer Panel 1 in the revised manuscript page 8-10 as given below 

Panel 1: Government district health facility framework and description of facilities in the context of PSBI management

Sl. no Name of the Facility/ Institution Expected strength of health personnel Type of care provided Population covered Number of health facilities in the study area Number of health personnel in position during the study period for neonatal care

1 Accredited Social Health Activist ( ASHA) Sanctioned positions for the area-172

 Social mobilizer for promoting maternal and child health: ante-natal care; deliveries; immunization and sickness identification, counseling, 

door-to-door surveys for specific service recipients @One ASHA for approximately 1000 rural population Working from home

 172 

2 Sub-Centre

(SC) 3. Two ANMs 

4. one male multi-purpose health worker (MPW) Lower most health service delivery facility; provides basic antenatal care, immunization and treatment for minor illnesses 3000-5000 18 ANM: 29 

Male Multi Purpose Health worker -3 

3 Primary Health Centre (PHC) 5. Two qualified doctors (1-allopathic-MBBS and 1- Ayush Traditional medicine physician) 

6. One staff nurse 

7. One ANM

8. Other staff not directly involved with neonatal care: 1 pharmacist; 1 health educator and 2 health assistants PHC- first port of call to a qualified public sector doctor in rural areas for the sick and those who directly report or referred from Sub-Centres (5-6 SC fall under each PHC) for curative, preventive and promotive health care.

4-6 Bedded facility 20000-30000 3 PHC 1: Nagaljaat

Medical Officer: 0

Nurse: 1

ANM: 2

PHC 2: Kot

Medical Officer: 2

Nurse: 0

ANM:1

PHC 3: Uttawar 

Medical Officer: 2

Nurse: 0

ANM:1

4 Community Health Centre (CHC) 4. Five medical specialists (i.e., general surgeon, physician, gynecologist, anesthetist and pediatrician)

5. Four general duty officers (1 dental surgeon, 3 general medical officers 

6. Twenty one paramedical and support other staff. First referral unit for SCs and PHCs;

Facilities with one operation theater, X-ray machine, labour room and laboratory. 

Provides facilities for 24X7 obstetric care and specialist consultations

Beds: 20-30 80000-120000 3 CHC 1: Aurangabad

Specialist: 0

Medical Officers: 4

Nurse: 1

ANM: 3

CHC 2: Hodal

Specialist: 0

Medical Officers: 4

Nurse: 4

ANM: 4

CHC 3: Hathin 

Specialist: 0

Medical Officers: 5

Nurse: 4

ANM: 2

5 District Hospital (DH) Palwal district hospital is a 

200 bedded hospital.

The district hospital has 

3. Thirty seven specialists (including 3 paediatricians) 

4. One hundred and thirty two paramedical staffs ( including 90 Staff nurse) District Hospital is secondary referral level facility and provides comprehensive secondary health care services to the population in the district. District Hospital is expected to deliver Essential (Minimum Assured Services) and Desirable (which we should aspire to achieve) package of services. The services include OPD, indoor and emergency service. In addition, basic specialty services, newborn care, psychiatric services, physical medicine and rehabilitation services, accident and trauma services, dialysis services and anti-retroviral therapy. Average population of a district varies between 15 to 30 million 

Palwal district hospital covers a population of 

1 million 1 

Palwal Civil Hospital

Paediatrician:3

Medical officers:6

Nurse: 6

A deputy Chief Medical Officer [Deputy-CMO] supervised ANMs and ASHA coordinators [supervisors for ASHAs] 

Health administration in the district managed by Chief Medical Officer’s Office

6 Special Newborn Care Unit (SNCU) Managed by an adequately trained pediatrician, doctors, staff nurses and support staff to provide 24x7 services.

As per MOHFW, the expected staff strength for a SNCU should be

3. One doctor per 4 beds 

4. One nurse per 1.5 beds

( Palwal SNCU should have 5 doctors and 12 nurses) A SNCU is established at the district hospital and sub-district hospitals to provide care for sick newborns, i.e., all type of neonatal care except assisted ventilation and major surgeries. It is a separate unit close to the labour room with 12 or more beds.

Palwal DH- SNCU has 18 beds to manage sick newborns >3000 to 20000 annual delivery at district /sub-district level 1 Same as mentioned in row 5 under the district hospital

28. Study design: Section should specify which implementation research principles were used and justify the selection. Generally, this section, the steps/methods used & how they map onto the objectives was difficult to follow. Would strongly suggest including a diagram or figure that maps both the steps of the process and the pieces of the formative evaluation onto the overall study objectives.

• Response: We have incorporated the study framework adopted from the RE-AIM framework. We have added the following sentence in the revised manuscript on page 10, line 184-185.

“RE-AIM framework [31] was adopted to evaluate the implementation of PSBI guidelines in the study (Fig 2).”

 Ref - Glasgow RE, Vogt TM, Boles SM. Evaluating the public health impact of health promotion interventions: the RE-AIM framework. Am J Public Health (1999) 89(9):1322–7. doi:10.2105/AJPH.89.9.1322

29. Details about the study protocols—especially for interviews, surveys, etc—are almost completely absent. Protocols for each of these data collection efforts should be (at least briefly) described, in a table or elsewhere. Also, this section consistently conflates research methods with results—for example, reporting final Ns for data collection instruments, rather than study protocols and procedures for recruiting individuals to complete measures, instruments, or interviews.

• Response: We have now provided study protocol in details. Study procedures data collection, management and analysis section has been modified now. 

Please refer line 257-291 in page s 14, 15 1ns 16 for protocol of formative phase 

“A formative study was conducted at baseline between June 2017 and January 2018. The assessment was done in two parts. 

Formative research-part 1(June-August 2017): Baseline assessment included four components. i) Facility assessment: WHO facility scoring tool [33] was adapted to assess readiness for the availability of PSBI related supplies, manpower, service delivery and mechanisms of monitoring implementation of PSBI guideline (DH-1, CHC-3, PHC-3, sub-centers-18). ii) ASHAs’ perspectives about HBNC program, home visitation, sickness identification among young infants and care provision were obtained through knowledge attitude and practice (KAP) survey (n- 60; at the rate of at least one ASHA per village). iii) Perspectives of recently delivered (<6 months) and lactating mothers: In-depth interviews (n-30 mothers) and KAP survey with 150 mothers (at the rate of 3 randomly selected mothers from every village) to document mothers’ awareness of danger signs, care-seeking for their sick newborn and perceived value of home visitation by the ASHAs. iv) Towards the end of part 1 of the formative phase, both in-depth interviews and the quantitative survey indicated ground-level challenges about home visitation under HBNC by the ASHAs, assessment and management of young infants at the PHCs and CHCs and its influence on the care-seeking by the families for sick young infants. The respondents (surveys of mothers and ASHAs) had touched on these issues but in a restrained manner probably due to the sensitive nature of the problems. Non-formal interactions (NFI) [34, 35, 36] with primary care physicians working at CHCs and PHCs (n-5), pediatric specialists at district hospital (n-2), ASHAs (n-8), ANMs (n-5) and mothers (n-11) were undertaken to further explore these issues. Only senior investigators conducted NFIs which were considered critical to the success of the implementation research. No tape recording or notes were taken at the time of interaction but immediately after NFI, the summary of the interaction was noted down. 

Co-participatory formative research – part 2 (August 2017- January 2018): The purpose was to get first-hand insight into challenges and barriers to implementing PSBI guideline beyond that obtained under Part 1 of the formative phase. The co-participant implementation phase was limited to six villages (population-29000); served by 71 ASHAs, two SCs, one CHC and the district hospital. During this period the research team also undertook house to house survey to determine the pregnancies missed by the health system (ASHAs), tracked them for deliveries and called up families with newborn to verify the ASHAs’ scheduled visits (For every new delivery a schedule of expected home visits by the ASHA was made and research staff called up the families between 48 and 96 hours after the scheduled time to confirm the home visits.) At least one of the investigators observed the doctors assessing infants aged less than 59 days at the CHC (receiving patients from the above mentioned six villages) using a structured observation checklist ( Supporting Information S1.Checklist) . This component was part of a planned staged implantation scale-up of the study; after the end of the six months, the study was expanded to all 50 villages”. 

Refer line no -359 to 387 in page 18 -19 describe data ccollection and management in details 

“The research data collection team comprised of Indian traditional medicine graduates with master’s in public health, graduate social scientists and field workers with at least 10 years of schooling. The research personnel were trained as per their assigned tasks in a 3-5 days’ workshop. They collected data for during the formative research; medical graduates interacted with medical officers in primary health facilities in the first two quarters, conducted post-treatment follow up (on 8th and 14th day) of PSBI diagnosis irrespective of the place where they were treated and abstracted administrative data from official documents of ASHAs, ANMs, PHCs, CHCs and district hospital. The data included: pregnancy, live birth, home visitation by ASHAs, sick young infants assessed and managed at different public sector health facilities, PSBI case classification formats, drugs administered, place of management (outpatient or in-patient), adherence, and follow up on day four and seven after initiating the treatment from all government health facilities. 

As part of the community mobilization and engagement activities, the families were asked to inform the research team telephonically if they chose to take their sick young infants to private providers (Table S1). Data about obtaining treatment at private health facilities, clinical features at the time of seeking care and outcome was based on the recall by the mothers and families and available prescriptions. Information was obtained by the research team on the 8th and 14th day follow up post-initiation of treatment. The information about the case classification was not available for these cases. The processes and experiences of different aspects of implementation were recorded in the daily diaries of the study investigator and team leaders.

Quantitative data (KAP surveys of mothers and ASHAs, facility assessments, pregnancy and birth surveillance, assessment of sick young infants, treatment provided and outcome) was double entered in RedCap® double data module, for detecting inconsistencies and merged after validation. Data was exported to STATA® (Version 15.0) for analysis. Descriptive statistics were used to present coverage data, treatment adherence and cross-sectional survey data. 

Qualitative data: Transcripts of recorded interviews (IDIs conducted during formative research with health service personnel, mothers and ASHAs) were prepared and complemented with field notes taken during non-formal interactions. All transcripts were entered in IQDAS (INCLEN Qualitative Data Analysis Software) [37]. Data were free-listed and key axial and selective codes were generated for analysis.” 

30. Policy dialogue: Panel 2 is interesting and clearly very important, but no context is given for why certain departures from the WHO guidelines were determined and/or how they were justified. This is very important to understand as implementation efforts assume implementation of an evidence-based practice; if these guidelines are forcing departure from that evidence-base, it undermines the entire endeavor. Alternatively, tailoring that is done to address known barriers is an important part of the implementation process and thus provision of details about why these changes were made should be provided.

• Response: We want the reiterate that the WHO guideline (2015) was used as an intervention package and not the GOI (2014) guideline. Thus we have used evidenced-based guideline only. In the section “Built coalition” we have mentioned that we organized a national workshop to harmonize GOI guideline with WHO guideline. There was no departure from the WHO guideline during implementation. 

Please refer line 207 – 213 in page 11

 “As part of the policy dialogue with both central and state ministries of health, the research team obtained permission for modification of the GOI PSBI management guideline in a national workshop (October 2016) attended by national and state program managers, independent experts and brought it in line with the WHO PSBI guideline specifically for the implementation research study (Panel 2). The same group of stakeholders also participated in the finalization of the research protocol. The research team provided technical support to the Haryana State NHM to modify the IMNCI recording form and PSBI case follow up card to manage and maintain follow-up records of PSBI infants treated in PHCs/CHCs.”

31. Implementation phase: Specifics on the implementation strategy used should be provided ideally in line with Proctor, Powell & McMillen (2013) and/or as specified through the ERIC classification of implementation strategies (cf. Powell et al, 2015).

• Response: We have modified the nomenclature of implementation strategies in the methodology section according to the ERIC classification. 

Please refer line no 194 – 195 in page 11

“To contextualize our implementation of the PSBI management intervention package, following discrete ERIC classification implementation strategies [32] were followed.”

32. Table 1: This table is clearly both very important and also incredibly onerous and almost entirely ineffective at communicating information. Additionally, the information included here seems to be related to results, not methods—so it should be moved, and the results should be presented in a more reader-accessible and organized way. One option may be to keep this table as supplemental information and highlight key results (in the results section!) that are curated by the authors as highlighting key lessons learned.

• Response: The table has now been moved to the result section. And text has been modified for more clearity. Please refer page 21-24

33. Nudges: No rationale or justification is provided for using the nudge strategy—which is notable because evidence regarding the effectiveness of nudges in changing behaviors is mixed. Additionally, it is not clear where or at what stages nudges were developed, how they were integrated into the overall implementation strategy, what protocol for nudges were & whether these protocols were followed with fidelity, and/or what results in they were expected to have. Finally, no citations are provided for definitions of or literature around behavioral nudges.

• Response: Thank you for this insight. We realized that what we leveraged as an implementation strategy is not a “behavioral nudge” in its true sense. We thus had no scope to measure the impact of behavioral nudge. We have now mentioned implementation strategies as per ERIC nomenclature. For the study, we did not develop a strict nudge protocol neither did we document the fidelity. 

34. Data collection/management: Again, results are presented here. No Ns should be presented in methods sections (other than population N or target Ns).

• Response: Now the section is modified as suggested. 

Refer line no -359 to 387 in page 18 -19 describe data ccollection and management in details 

“The research data collection team comprised of Indian traditional medicine graduates with master’s in public health, graduate social scientists and field workers with at least 10 years of schooling. The research personnel were trained as per their assigned tasks in a 3-5 days’ workshop. They collected data for during the formative research; medical graduates interacted with medical officers in primary health facilities in the first two quarters, conducted post-treatment follow up (on 8th and 14th day) of PSBI diagnosis irrespective of the place where they were treated and abstracted administrative data from official documents of ASHAs, ANMs, PHCs, CHCs and district hospital. The data included: pregnancy, live birth, home visitation by ASHAs, sick young infants assessed and managed at different public sector health facilities, PSBI case classification formats, drugs administered, place of management (outpatient or in-patient), adherence, and follow up on day four and seven after initiating the treatment from all government health facilities. 

As part of the community mobilization and engagement activities, the families were asked to inform the research team telephonically if they chose to take their sick young infants to private providers (Table S1). Data about obtaining treatment at private health facilities, clinical features at the time of seeking care and outcome was based on the recall by the mothers and families and available prescriptions. Information was obtained by the research team on the 8th and 14th day follow up post-initiation of treatment. The information about the case classification was not available for these cases. The processes and experiences of different aspects of implementation were recorded in the daily diaries of the study investigator and team leaders.

Quantitative data (KAP surveys of mothers and ASHAs, facility assessments, pregnancy and birth surveillance, assessment of sick young infants, treatment provided and outcome) was double entered in RedCap® double data module, for detecting inconsistencies and merged after validation. Data was exported to STATA® (Version 15.0) for analysis. Descriptive statistics were used to present coverage data, treatment adherence and cross-sectional survey data. 

Qualitative data: Transcripts of recorded interviews (IDIs conducted during formative research with health service personnel, mothers and ASHAs) were prepared and complemented with field notes taken during non-formal interactions. All transcripts were entered in IQDAS (INCLEN Qualitative Data Analysis Software) [37]. Data were free-listed and key axial and selective codes were generated for analysis.” 

Result

35. As with the methods section, it is difficult to follow which results track from which stages of the process. I would again encourage picking a structure for presenting the overall study design (e.g., in phases), mapping the objectives/research questions to those phases, and then mapping results (in this section) to those phases/questions. As presented currently, the conflation of methods for evaluation and results of those evaluations are very convoluted and difficult to extricate to determine whether results are valid or reliable, or whether they answer questions of interest.

• Response: In the study design section, the overall study design is presented in a phased manner. We have attempted to differentiate between implementation and research (evaluation) in the framework itself. The results are now separated from the methods. 

 Please refer aour contextualized Re AIM framework figure in page 10 line 191 wher in we have presented study design in phase manner . 

36. Table 2: Unclear whether the days listed in Column 1 are ranges (e.g., is the second line reflecting visits that occurred between Day 1 and Day 3 or just on Day 3)? Also unclear why 0 visits is inapplicable to the partial information category. Finally, I didn’t see the response rate indicated anywhere in the table or text—is it correct that the response rate is (2001+487)/16,997 total calls, i.e., less than 15%?

• Response: The table number is now table 3. The data presented in Column 1 are not ranges. Each line representing the visits on that particular day. The partial information category is now deleted to avoid confusion. 

Refer page 28 

Table 3. Post-natal home visitation by ASHAs during first four quarters (Aug 2017 - Jul 2018) of study period *

Post-natal home visits by ASHAs Based on the official record submitted by ASHA

(N – 1475) † Based on the call from the research team to families

(complete information available about the scheduled visits)

(N – 2001) ‡ 

 n (%) n (%)

• Day 1 366(24.8) 372(18.6)

• Day 3 1341(90.9) 1079(53.9)

• Day 7 1414(95.9) 923(46.1)

• Day 14 1385(93.9) 880(44.0)

• Day 21 1324(89.8) 799(40.0)

• Day 28 1250(84.7) 785(39.2)

• Day 42 1168(79.2) 854(42.7)

• 0 visits 0 (0%) 336(16.8)

*Total no. of live births during Aug 2017-Jul 2018 – 3,254

†Refusal of ASHA to share records of 1,779 mother–child dyads

‡Total no. of calls made to the families - 16,977

37. Table 3: The presentation of this table (as well as some of the text in the conclusion) suggests that there were some over-time improvements in implementation, however, these effects are not apparent from this table—perhaps because there is so much information that is presented here? One suggestion would be to curate this table a bit more to focus on the results that were most interesting? For example, (a) since sub-centres were never the first point of care nor place of treatment, these rows could be removed from the table & a footnote could indicate this absence of action. Additionally, (b) if changes over time are of interest, proper statistical tests should be included. (Note (c) that a multi-paneled figure reflecting trends would be a much more reader-friendly way to convey this information as well).

• Response : 

(a) Sub-centre row is now deleted.

(b) Multi paneled figure to show the change is now attached as Supporting information .

PSBI management:

38. Unclear where the 10% live births with PSBI assumption comes from; this should be justified and sources referenced.

• Response: This data is referenced now.

Please refer line 427 in page 26

Saha SK, Schrag SJ, El Arifeen S, Mullany LC, Shahidul Islam M, Shang N, et al. Causes and incidence of community-acquired serious infections among young children in south Asia (ANISA): an observational cohort study. The Lancet. 2018;392:145–59.

African Neonatal Sepsis Trial (AFRINEST) group, Tshefu A, Lokangaka A, et al. Simplified antibiotic regimens compared with injectable procaine benzylpenicillin plus gentamicin for treatment of neonates and young infants with clinical signs of possible serious bacterial infection when referral is not possible: a randomised, open-label, equivalence trial. Lancet 2015; 385: 1767–76 pmid:25842221

Mir F, Nisar I, Tikmani SS, Baloch B, Shakoor S, Jehan F et al. Simplified antibiotic regimens for treatment of clinical severe infection in the outpatient setting when referral is not possible for young infants in Pakistan (Simplified Antibiotic Therapy Trial [SATT]): a randomised, open-label, equivalence trial. Lancet Glob Health 2017; 5: e177–85 pmid:27988146

Baqui A, Saha S, Ahmed A, Shahidulla M, Quasem I, Roth DE et al. Safety and efficacy of alternative antibiotic regimens compared with 7 day injectable procaine benzylpenicillin and gentamicin for outpatient treatment of neonates and young infants with clinical signs of severe infection when referral is not possible: a randomised open label, equivalence trial. Lancet Glob Heal. 2015; 3: e279–87

39. Table 4: Best practices in this table should be highlighted to indicate what proportion of cases were ‘high fidelity’. Additionally, having a separate category for 6 cases (Critical illness) is problematic—would combine with one of the other categories (or simply not include as its own column). I also don’t think sections B1 and B2 need to be in the table since they are not a focus of the implementation effort. Discuss in the text but exclude from the table.

• Response : 

i. Row number 2 under section A represents the high fidelity cases.

ii. We belive that the ‘Critical Illness’ should be represented separately, as they are seriously sick infants, who ideally should not be managed in primary facilities; However, because they refused hospitalization they were managed at the primary care facilities. This is a high risk for mortality category that cannot be combined with other categories. 

iii. Our aim was also to incarese access to treatment for PSBI also. Management of PSBI at the referral level facilties is important because this is the current standard of care. That is why we included this information. We felt this is important to mention and it gives a complete picture. 

iv. We have separated the ‘Private Providers’ from the ‘Hospital Management’ as ‘C’. The reviewer 1 recommended the private sector data to be presented. l

40. Unclear what the purpose of is including section on sicknesses identified other than PSBI—this should be dropped, and other relevant information related to barriers to care should be expanded. 

• Response: Again, other illnesses were presented to complete the picture. We realized that, to make a health facility functional (in terms of managing sick young infants), one should include other illnesses alongwith the PSBI cases. It is crucial to build trust of the community in order to influence their careseeking behaviour in a positive manner.

Discussion

41. Unclear where the 70% treatment coverage rate came from—this should be defined precisely in the results section.

Response: We have clarified the language about treatment coverage. Previously published data reported around 10% prevalence of PSBI in young infants up to 2 months of age (see references below). We used this 10% for calculating the coverage purpose. Considering 5270 live births in our study area during the study tenure, our estimated cases would have been 527, but in our study area we identified only 370 cases that were treated, which indicates a coverage of 70%. 

Saha SK, Schrag SJ, El Arifeen S, Mullany LC, Shahidul Islam M, Shang N, et al. Causes and incidence of community-acquired serious infections among young children in south

 Asia (ANISA): an observational cohort study. The Lancet. 2018;392:145–59.

African Neonatal Sepsis Trial (AFRINEST) group, Tshefu A, Lokangaka A, et al. Simplified antibiotic regimens compared with injectable procaine benzylpenicillin plus gentamicin for treatment of neonates and young infants with clinical signs of possible serious bacterial infection when referral is not possible: a randomised, open-label, equivalence trial. Lancet 2015; 385: 1767–76 pmid:25842221

Mir F, Nisar I, Tikmani SS, Baloch B, Shakoor S, Jehan F et al. Simplified antibiotic regimens for treatment of clinical severe infection in the outpatient setting when referral is not possible for young infants in Pakistan (Simplified Antibiotic Therapy Trial [SATT]): a randomised, open-label, equivalence trial. Lancet Glob Health 2017; 5: e177–85 pmid:27988146

42. Baqui A, Saha S, Ahmed A, Shahidulla M, Quasem I, Roth DE et al. Safety and efficacy of alternative antibiotic regimens compared with 7 day injectable procaine benzylpenicillin and gentamicin for outpatient treatment of neonates and young infants with clinical signs of severe infection when referral is not possible: a randomised open label, equivalence trial. Lancet Glob Heal. 2015; 3: e279–87Generally, conclusions need to be tempered for this paper, and the conclusions should remind the readers from the outset that this was an observational, feasibility study for implementation of a particular guideline. While this paper does seem to strongly support feasibility of this intervention, there are serious limitations related to the selection of the region, problems with data reporting, and of course the study design that limit generalizability of findings, and certainly limit any notions of effectiveness or causality of the implementation. This is particularly notable for the authors claim that this study “demonstrated that oral amoxicillin alone is effective in managing pneumonia cases”—such a claim simply cannot be made with a study like this as there is no control case; data is purely observational. Further, claims like this distract from the implementation focus of the study. The authors would be much better off focusing on the purview suggested by the title—lessons from implementation—rather than trying to make claims related to clinical effectiveness. Of course, this doesn’t mean that they cannot claim that low mortality rates and high treatment rates aren’t important; they are—but they are reflective of the potential for guidelines like this to be implemented in communities like this that have room for improvement; their “effect” on downstream clinical outcomes simply cannot be estimated without a more rigorous study design.

• Response: We have modified the conclusion section as per the suggestions 

Please refer line 617- 626 in page 37

“In conclusion, we identified some key implementation barriers such as irregular and poor quality home visitation by the ASHAs, poor ability of the families and mothers to identify danger signs in their young infants, lack of confidence of primary care physicians at PHCS/CHCs to manage sick young infants and reluctance of the ANMs to consider themselves as care providers. Most of the bottlenecks could be resolved by the district and state authorities and technical assistance from experts through leveraging existing resources and developing contextualized strategies. We demonstrated the feasibility of implementing management of PSBI on an outpatient basis when referral to a hospital was not feasible in a program setting by focusing on hand holding and confidence building of the primary care physicians, making the CHCs and PHCs functional and targeted social mobilization to empower the mothers and families for early recognition of sickness in their young infants”. 

43. As with the rest of the manuscript, the conclusion has several places where important references appear to be missing—e.g., for “recent implementation research” that describes adherence to antibiotics guidelines improving slowly over time.

• Response: We have removed the statement. 

44. The limitations should reflect the limitations of the observational study design, potential threats to validity and generalizability related to the population of interest, and larger issues related to data quality and potential missing data. This section is very weak at the moment.

• Response : Section has been modified. 

Please refer line 601- 614 in page 36

The results from our observational feasibility study need to be seen in light of its limitations. We were able to identify a much smaller number of infants with fast breathing due to a concurrent study about the management of the fast breathers by ASHAs, which led to relatively lower coverage of the sick young infants. But it is unlikely to influence the documentation of our experience of using simplified antibiotic including injectable antibiotics for other more severe categories of PSBI. Home visits by ASHAs were irregular and of poor quality for empowering the families and mothers, which led to a relatively small proportion of sick young infants by them. ASHAs are permanent member of the public health field team and can provide the sustainability to PSBI program, so the district authorities need to follow-up to increase the postnatal home visits. We did not cross-check the antibiotic doses administrated by the public facility doctors. Thus we could only comment on the effectiveness of simplified antibiotic therapy but cannot comment on the drug dosage and its response. Finally, for various operational and administrative reasons, we had to terminate the study without executing the panned exit strategy prepared in consultation with the TSU, which is likely to influence the sustainability of program refinements after the exit of the research team. 

---

## [Decision Letter · Decision Letter 1]

21 May 2021

Lessons from an implementation research on community management of possible serious bacterial infection (PSBI) in young infants (0-59 days), when referral is not feasible in Palwal district of Haryana, India.

PONE-D-20-18867R1

Dear Dr. Arora,

We’re pleased to inform you that your manuscript has been judged scientifically suitable for publication and will be formally accepted for publication once it meets all outstanding technical requirements.

Kind regards,

Ricardo Q. Gurgel, PhD

Academic Editor

PLOS ONE

Additional Editor Comments (optional):

Reviewers' comments:

Reviewer's Responses to Questions

**Comments to the Author**

1. If the authors have adequately addressed your comments raised in a previous round of review and you feel that this manuscript is now acceptable for publication, you may indicate that here to bypass the “Comments to the Author” section, enter your conflict of interest statement in the “Confidential to Editor” section, and submit your "Accept" recommendation.

Reviewer #1: All comments have been addressed

Reviewer #3: (No Response)

Reviewer #4: (No Response)

2. Is the manuscript technically sound, and do the data support the conclusions?

Reviewer #1: Partly

Reviewer #3: Yes

Reviewer #4: Partly

3. Has the statistical analysis been performed appropriately and rigorously? 

Reviewer #1: Yes

Reviewer #3: Yes

Reviewer #4: Yes

4. Have the authors made all data underlying the findings in their manuscript fully available?

Reviewer #1: Yes

Reviewer #3: Yes

Reviewer #4: Yes

5. Is the manuscript presented in an intelligible fashion and written in standard English?

Reviewer #1: Yes

Reviewer #3: Yes

Reviewer #4: Yes

6. Review Comments to the Author

Reviewer #1: (No Response)

Reviewer #3: This is an important paper relates to WHO guideline on implementation of management of possible severe bacterial infection in low-middle income countries that has critical importance for global public health and findings of the study could affect policy and practice in other settings. It is professionally written paper. I do not have major reservations. However, the paper needs minor revision on the following issues before it can be considered for publication.

Please delete lines 247-252 “out of family…………………… S1 table” from method section and author may describe it is result section.

Table 2 can be deleted and replace in brief narration.

Please revise Table 1 as it is too lengthy. It would be rendered concise and comprehensive for the ease of readers.

There is some repetition in the paper.

Please delete lines 303-307 “there were two …………………………………………………… in the CHCs and PHCs (Table 1)”. As it is repetition, already documented in table 1.

Please delete line 316-325 “Implementation of nudge strategies” section as it is repetition.

Reviewer #4: 1. This research should be viewed in the context of a series of studies conducted in different geographical locations in India that have been published in your journal,

PLOS ONE. This is a part of the implementation research on WHO PSBI guidelines.

2. The objective of the study is to implement the PSBI guideline developed by the WHO & GOI

3. The RE- AIM Framework was adopted to document the implementation strategy.

4. Chance of working in a real world setting, strength and weaknesses of different approaches have been tested using the Reach, Effectiveness, Adoption, Implementation & Maintenance approaches that are together expected to determine the Public Health Impact of the intervention.

5. Considering the above, the individual level impact in terms of reaching the target population and efficacy of the intervention have been achieved in good measure in the study ( 70% coverage and above 90% achievement of clinical treatment success). However, a higher Coverage of intervention (80%) was achieved in Himachal Pradesh (Goyal et al). Identification of PSBI by ASHA workers was reported to be 80% in the study by Awasthi et al in Uttar Pradesh.

As members of this collaborative research team, the authors of the manuscript could have used learnings from these settings to improve the implementation in Palwal.

6. One can not comment on the consistency and the cost of intervention implementation, as it has not been evaluated or reported in the manuscript.

7. Maintenance of the intervention effects in the study setting was probably not possible to observe, as the authors report termination of the study without executing the planned exit strategy, under limitations.

8. The TSU served as the backbone in the implementation research study: although the strategies to empower mothers and families, building capacity through training of health staff seems to be partially successful, the strategy for improving performance of the ASHAs, ANMs seems to have failed.

It is mentioned that the health system laid emphasis on the registration /line listing of the beneficiaries by the ASHAs, however, completing the visitation as per schedule of HBNC was not prioritised.

The authors have not explained any reasons for this. It is mentioned that Non formal interactions with health care providers were conducted to explore these issues in a restrained manner. However, better remedial measures if implemented under the guidance of the TSU/ or on the basis of feedback from the research team that co-participated in the implementation for initial six months to identify barriers, could have led to improved coverage of HBNC visits by ASHA resulting in better identification of PSBI as well.

Learnings from the efficacy trials such as the Gadchiroli study (Bang et al The Lancet 1999) and more recently results of an M-health intervention study conducted in Gujarat (Modi et al PLOS One- 2019) indicate that Village health workers now called ASHAs, if trained adequately can help improve the morbidity mortality indicators immensely.

9. The qualitative research has been designed and implemented well. Community mobilization activities have been described well.

10. Data presented in Table 2 describes identification, point of care, place of treatment, number of deaths by six Quarters, however, no trends can be observed in the data presented. It is not helpful in understanding the impact of study activities on outcomes.

11. Table 3 Post natal Home visitation by ASHAs shows substantial discrepancy in data reported by ASHA and the data collected by the research team on phone calls: The reasons for this should have been better discussed, it is just mentioned that incomplete data on mother-infant dyad was not reported.

12. The ASHAs identified only one tenth of the PSBI cases: was the incentive based payment system working adequately in the study setting? could there be monetary reasons for underperformance among others, listed by the authors in the discussion section.?

13. Similarly, what were the reasons for better impact on mothers and family members as compared to ASHAs and other health workers?

14.The Implementation research study identified the bottlenecks of implementation of PBSI guideline in presence of a TSU and its positive role, well emphasised in the manuscript. The reasons for failure of its strategies to improve service delivery by ASHAs, ANMs needs to be further explored

7. PLOS authors have the option to publish the peer review history of their article (what does this mean?). If published, this will include your full peer review and any attached files.

Reviewer #1: No

Reviewer #3: **Yes: **Prof Sajid Soofi

Reviewer #4: **Yes: **Anju Sinha

---

## [Editor Report · Acceptance letter]

23 Jun 2021

PONE-D-20-18867R1 

Lessons from implementation research on community management of possible serious bacterial infection (PSBI) in young infants (0-59 days), when the referral is not feasible in Palwal district of Haryana, India. 

Dear Dr. Arora:

I'm pleased to inform you that your manuscript has been deemed suitable for publication in PLOS ONE. Congratulations! Your manuscript is now with our production department. 

Kind regards, 

on behalf of

Professor Ricardo Q. Gurgel 

Academic Editor

PLOS ONE